# 2′-*O*-methylation-dependent installation of *N²*-methylguanosine in the U6 internal stem loop facilitates efficient spliceosome assembly

Nicole Kleiber [1], Jonny Petrosyan[1], Maria Greve[2], Chairini C. Thomé[1], Olexandr Dybkov[3], Laurianne L. E. Tay [4,5], Luisa M. Welp[3,6], Philipp Hackert[1], David Touboul [5], Holger Stark[7], Markus T. Bohnsack[1,8], Henning Urlaub [3,6], Lydia Herzel [9], Marc Graille [4], Claudia Höbartner [2,10] & Katherine E. Bohnsack [1,8] ✉

The internal stem loop (ISL) of the human U6 snRNA, which catalyzes pre-mRNA splicing, contains LARP7-dependent, snoRNA-guided 2′-*O*-methylations and an *N²*-methylguanosine (m²G) that is required for splicing of weak splice sites. Here, we show that installation of m²G$_{72}$ by the THUMPD2-TRMT112 methyltransferase complex is one of the last maturation events during U6 snRNP biogenesis. We dissect features of THUMPD2 required for association with U6 and present an experimentally validated model of the THUMPD2-TRMT112-U6 complex. Using in vitro methylation assays as well as a newly developed m²G-sensitive deoxyribozyme to monitor U6-m²G$_{72}$ levels in cellular RNAs, we reveal that 2′-*O*-methylations within the U6 ISL enhance methylation of G$_{72}$. We show that m²G$_{72}$ and the 2′-*O*-methylations in U6 independently and interdependently influence alternative splicing. Furthermore, our data demonstrate that 2′-*O*-methylations in the ISL are required for incorporation of U6 into snRNPs whereas m²G$_{72}$ influences the progression of the U6 snRNP into larger assemblies, highlighting distinct roles of these modifications during spliceosome assembly.

Modified nucleotides are present throughout the transcriptome where they fulfill important roles in fine-tuning and regulating gene expression[1–5]. The RNA modification landscape is formed by numerous enzymes that modify the four standard nucleotides (adenosine (A), uridine (U), cytidine (C) and guanosine (G)) in diverse ways, including methylating different positions of the bases and 2′-*O*-methylation of the ribose moieties[6–8]. Non-coding RNAs, such as small nuclear RNA (snRNAs), are densely populated with modified nucleotides that

[1]Department of Molecular Biology, University Medical Center Göttingen, Justus-von-Liebig-Weg 11, Göttingen, Germany. [2]Institute of Organic Chemistry, Universität Würzburg, Am Hubland, Würzburg, Germany. [3]Max Planck Institute for Multidisciplinary Sciences, Research Group 'Bioanalytical Mass Spectrometry', Am Fassberg 11, Göttingen, Germany. [4]Laboratoire de Biologie Structurale de la Cellule (BIOC), CNRS, Ecole polytechnique, Institut Polytechnique de Paris, Palaiseau, France. [5]Laboratoire de Chimie Moleculaire (LCM), CNRS, Ecole polytechnique, Institut Polytechnique de Paris, Palaiseau, France. [6]Department of Clinical Chemistry, University Medical Center Göttingen, Robert-Koch-Straße 40, Göttingen, Germany. [7]Max Planck Institute for Multidisciplinary Sciences, Research Group 'Structural Dynamics', Am Fassberg 11, Göttingen, Germany. [8]Göttingen Center for Molecular Biosciences, Georg-August University Göttingen, Justus-von-Liebig-Weg 11, Göttingen, Germany. [9]Institute of Chemistry and Biochemistry, Freie Universität Berlin, Takustraße 6, Berlin, Germany. [10]Center for Nanosystems Chemistry, Julius-Maximilians-Universität Würzburg, Würzburg, Germany. ✉e-mail: katherine.bohnsack@med.uni-goettingen.de

enable them to act as stable scaffolds within ribonucleoprotein complexes (RNPs) and optimize base-pairing interactions[2–4,9–11]. Due to the important roles of RNA modifications in ensuring the fidelity of gene expression, defects in the epitranscriptome often have pathological consequences[12–14].

Alongside RNA modifications, another key step in the maturation of most human mRNAs is the removal of introns by pre-mRNA splicing; 98% of human genes contain introns and alternative splicing allows to produce multiple proteins from a single gene[15,16]. A series of spliceosomal complexes assembled de novo on each pre-mRNA substrate drive intron removal in two trans-esterification reactions[15,17]. The snRNAs of the major (U1, U2, U4, U5 and U6) and minor (U4atac, U5, U6atac, U11 and U12) spliceosomes base-pair with each other and conserved sequences in introns (5′ splice site (SS), 3′ SS, branch-point, etc.) to position the ribozymal U6/U6atac snRNAs appropriately for catalysis[3,15,17,18]. The U6 snRNP forms a di-snRNP with U4 (U4/U6), before joining of the U5 snRNP generates the tri-snRNP complex (U4/U6•U5) that is recruited to introns already associated with U1 and U2. Upon structural rearrangements, the internal stem-loop of U6 forms the catalytic core of active spliceosomal complexes and chelates the two magnesium ions necessary for branching and exon ligation[19,20]. The efficiency and fidelity of pre-mRNA splicing is ensured by a plethora of snRNA modifications that facilitate the establishment of essential architectural features of snRNP complexes and fine-tune snRNA base-pairing interactions[3,9].

The Sm-class snRNAs (e.g., U1, U2, U4, U5), synthesized by RNA polymerase (Pol) II, are modified with an $m^7G$ cap and exported to the cytosol[21,22] where they are bound by seven Sm proteins, the cap is hypermethylated to an $N^2$-7-trimethylated cap ($m^{2,2,7}G$ or $m_3G$) and the 3′ ends are trimmed[21,23]. Upon re-import into the nucleus, Sm-class snRNAs transiently accumulate in Cajal bodies where they undergo further maturation steps, including the modification of various nucleotides[3,24]. By contrast, the Lsm-class U6 and U6atac snRNAs are synthesized by Pol III, and as such, have heterogenous polyuridine 3′ ends that are bound by the La chaperone[25,26]. During their maturation, which involves addition of a monomethylphosphate 5′ cap by MEPCE, polyuridylation by TUT1 and the installation of small nucleolar RNA (snoRNA)-guided modifications, the U6/U6atac snRNAs pass through the nucleolus[25,27–29]. In the nucleoplasm, the 3′ end of the U6 snRNA is trimmed by USB1 leaving five uridines ending in a 2′,3′-cyclic phosphate[30]. U6 3′ end maturation triggers recruitment of the Lsm2-8 proteins that act as nuclear retention signals for U6 and facilitate association of the spliceosome-associated factor SART3[31,32]. Binding of SART3 promotes U6 ISL unwinding, enabling alternative base-pairing with U4 and the di-snRNP complex to be established[33,34]. SART3-bound di-snRNP complexes localize to Cajal bodies where U5 snRNPs associate, displacing SART3 and generating U4/U6•U5 tri-snRNPs complexes that are incorporated into pre-catalytic spliceosome B complexes[35,36]. Importantly, this dramatic structural rearrangement of the U6 ISL, which harbors the ribozyme active site, ensures that immature U6 snRNPs are catalytically inert and only become active upon correct assembly within $B^{act}$ complexes.

During their biogenesis, snRNAs are extensively 2′-O-methylated and pseudouridylated by snoRNPs (U6/U6atac) and small Cajal body-associated RNPs (scaRNPs; Sm-class snRNAs)[3,29,37]. The base-pairing of box C/D and H/ACA sno/scaRNAs with their snRNA target sites directs modification of specific nucleotides by the methyltransferase fibrillarin (FBL) and pseudouridine synthetase dyskerin (DKC), respectively[3,38]. In total, 20 2′-O-methylated nucleotides and 24 pseudouridines are present in the human snRNAs of the major spliceosome whereas only 2 and 4, respectively, are present in the minor spliceosomal snRNAs[3]. A handful of base methylations are also introduced into the major snRNAs by non-RNA-guided RNA methyltransferases. For example, METTL16 methylates N6 of $A_{43}$ of the U6 snRNA, and lack of this modification causes retention of introns containing an A at the

fourth position downstream of the 5′ SS, which base-pairs with U6-$m^6A_{43}$[39–44]. The U6 snRNA contains a second methylated base, $m^2G_{72}$, and the holoenzyme THUMPD2, which functions together with a cofactor TRMT112, was recently identified as the cognate methyltransferase[45,46]. Loss of THUMPD2 perturbs splicing with retained introns typically displaying lower splice site quality and shorter polypyrimidine tracts than unaffected introns[46]. The pre-mRNAs aberrantly spliced due to lack of U6-$m^2G_{72}$ are subjected to nonsense-mediated decay, leading to reduced mRNA levels[45,46]. Interestingly, reduced THUMPD2 expression is observed in age-related macular degeneration (AMD) and AMD-associated transcripts are abundant within those alternatively spliced in a THUMPD2-dependent manner[46]. While perturbation of snRNA modifications has been linked to regulation of splicing, how specific modified nucleotides impact spliceosome assembly and function remains largely unexplored. Although THUMPD2 was recognized as the enzyme responsible for installing the U6-$m^2G_{72}$ modification, it was unknown when during U6 biogenesis this methylation takes place, which features of THUMPD2 and the U6 snRNA are necessary for this event, and how, mechanistically, the presence of U6-$m^2G_{72}$ influences splicing.

## Results

### THUMPD2 crosslinks to the U6 snRNA in cells and is required for installation of $m^2G_{72}$

THUMPD2 has been recently characterized as an $m^2G$ methyltransferase and found to introduce $m^2G_{72}$ in the internal stem loop (ISL) of the human U6 snRNA[45,46]. However, $m^2G$ is present in other RNAs[8,47,48], so to obtain a transcriptome-wide perspective on RNAs directly associated with THUMPD2, UV crosslinking and analysis of cDNA (CRAC)[49,50] was performed. HEK293 cells over-expressing THUMPD2-His6-2xFLAG or the His6-2xFLAG tag were UV-crosslinked and protein–RNA complexes immunoprecipitated (Supplementary Fig. S1). Recovered RNA fragments were converted into a cDNA library that was deep sequenced. Mapping of the sequencing reads to the human genome revealed that among the reads in peaks significant in THUMPD2-His6-2xFLAG compared to the control, the vast majority were derived from snRNAs (Fig. 1a; and Supplementary Data 1). More specifically, up to 80% of the reads in the THUMPD2-His6-2xFLAG peaks arose from the U6 snRNA (Fig. 1a; and Supplementary Data 1). Consistent with previous data showing that lack of THUMPD2 does not alter $m^2G$ levels in poly(A)+ RNAs or tRNAs[46,51], these data indicate that U6 is the RNA predominantly bound by THUMPD2 in cells.

To demonstrate the specific requirement of THUMPD2 for the introduction of U6-$m^2G_{72}$, the levels of all methylated nucleosides present in the U6 snRNA ($m^6A$, Am, $m^2G$, 2′-O-methylated guanosine (Gm) and 2′-O-methylated cytidine (Cm)) were monitored by liquid chromatography high-resolution mass spectrometry (LC-HRMS) in U6 snRNA isolated from wild-type and THUMPD2 knockout (KO) cells[45]. With the exception of $m^2G$, which was detected in U6 snRNA from wild-type, but not THUMPD2 KO cells, the ratio of each of these modified nucleosides compared to unmodified equivalents was similar (Fig. 1b; and Supplementary Data 2). This demonstrates that THUMPD2 is required for installation of U6-$m^2G_{72}$ and that lack of this modification does not affect the methylation of any other nucleotides in U6.

### THUMPD2-TRMT112 associates with the U6 snRNP during the late stages of its biogenesis

Precursor U6 snRNA is synthesized with additional 3′ uridines and this tail is extended by TUT1 during early stages of U6 biogenesis before trimming by the USB1 ribonuclease[25,28,30]. As the timing of interaction of THUMPD2 with the U6 snRNA was unclear, 3′ rapid amplification of cDNA ends (3′ RACE) was performed to monitor the maturation status of U6 snRNA associated with THUMPD2. RNA co-immunoprecipitated with THUMPD2-His6-2xFLAG or the His6-2xFLAG tag was 3′ end

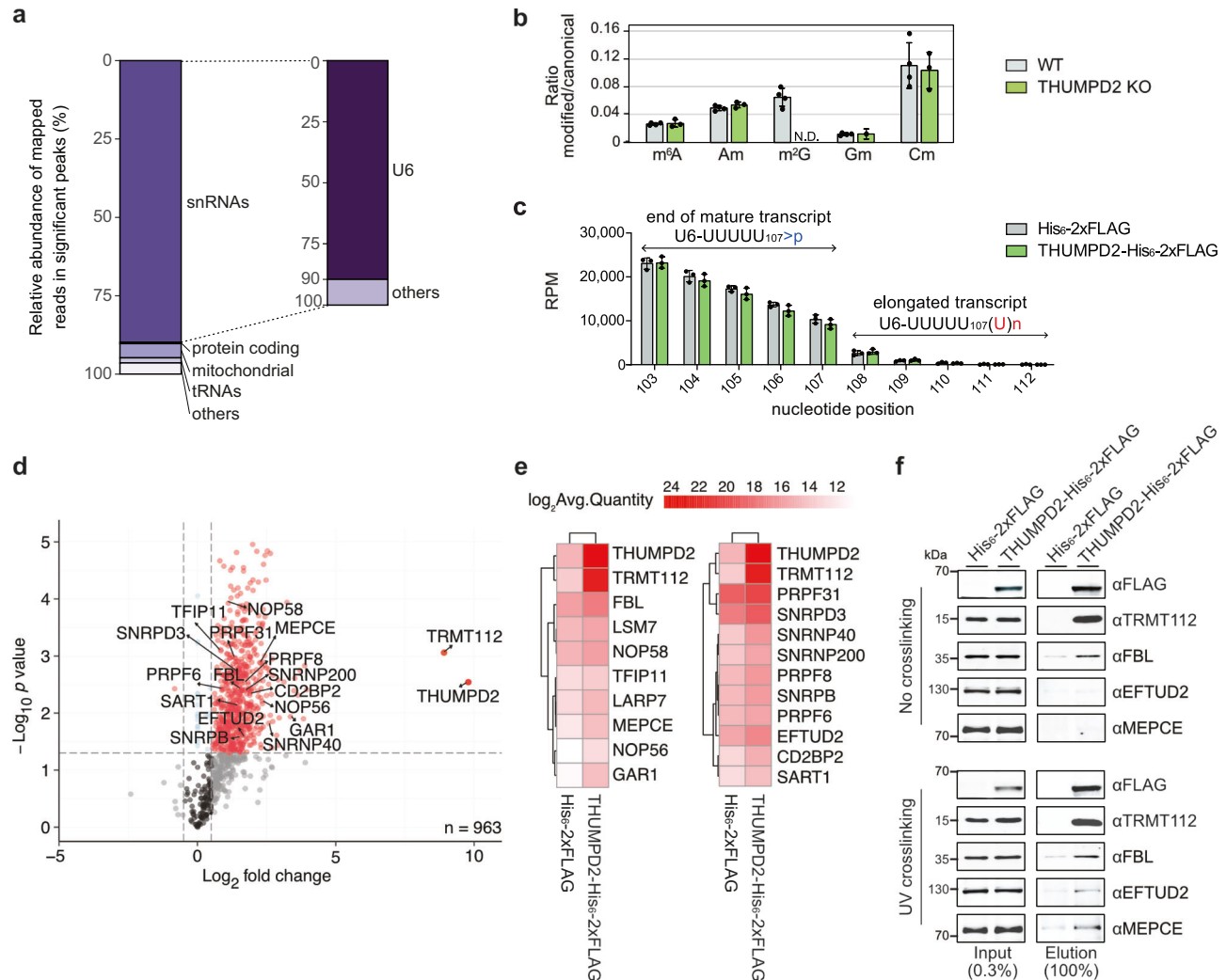

**Fig. 1 | THUMPD2 interacts with a processed form of the U6 snRNA and associates with the U6 snRNP during late biogenesis. a** HEK293 cells over-expressing His₆-2xFLAG or THUMPD2-His₆-2xFLAG were subjected to CRAC analysis to detect crosslinked RNAs. Stacked bar plots show the relative distribution of sequencing reads within significant peaks in $n = 2$ THUMPD2-His₆-2xFLAG datasets compared to His₆-2xFLAG controls. Data from one replicate is shown. A magnified view of the distribution of reads derived from snRNAs is also displayed. **b** LC-HRMS-based quantification of modified nucleotides in U6 snRNA purified from HCT116 WT (grey) or THUMPD2 KO (green) cell lines. Error bars represent mean ± standard deviation calculated from $n = 4$ (WT) or $n = 3$ (THUMPD2 KO) replicates (N.D. not detected). **c** RNAs co-immunoprecipitated with His₆-2xFLAG or THUMPD2-His₆-2xFLAG were subjected to 3′ RACE to map the 3′ ends of the U6 snRNA. Sequencing reads were aligned to the sequence of the oligouridylated U6 snRNA precursor and normalized by reads mapped per million reads (RPM) to allow the distribution of low-count reads in His₆-2xFLAG to be compared to THUMPD2-His₆-2xFLAG. The average numbers of normalized reads in $n = 3$ independent experiments mapping to each nucleotide of the precursor U6 snRNA 3′ end are shown as bar plots where

error bars represent mean ± standard deviation. **d** Lysates from UV-crosslinked HEK293 cells over-expressing His₆-2xFLAG or THUMPD2-His₆-2xFLAG were used for immunoprecipitation experiments. Eluates were nuclease-treated and proteins were detected by MS using data-independent acquisition. The differential abundance of peptides is shown as log₂ fold change (THUMPD2-His₆-2xFLAG vs. His₆-2xFLAG) against the significance level ($p$ values calculated using a Welch two-sided t-test) for $n = 4$ independent replicates. Nuclear proteins identified with a log₂ fold change >0.5 and $p$ value < 0.05 (indicated by the dotted lines) are colored in red. **e** Heatmaps showing the log₂-transformed MS2 intensity-based abundance of U6-associated proteins (U6 biogenesis factors (left) and splicing factors (right)) for proteins recovered in (**d**) together with THUMPD2-TRMT112. Splicing factors were searched against the Spliceosome Database[116]. **f** Immunoprecipitations with or without UV-crosslinking were performed as in (**d**) and input and eluate samples were analyzed by western blotting using the indicated antibodies. $n = 3$ experiments were performed and a representative image is shown. Source data and $p$ values are provided as a Source Data file.

dephosphorylated prior to library preparation and the obtained sequencing reads were mapped to the precursor U6 sequence. No increase in the numbers of reads mapping to the non-genomically encoded uridines was observed for THUMPD2-His₆-2xFLAG (Fig. 1c; and Supplementary Data 3), indicating that THUMPD2 interacts with mature U6, after processing of the polyuridylated 3′ end.

To further dissect when during snRNP biogenesis THUMPD2 associates with U6, immunoprecipitation (IP) experiments were performed with lysates from cells over-expressing THUMPD2-His₆-2xFLAG or His₆-2xFLAG. Cells were UV-crosslinked before IP to

facilitate the detection of protein–protein interactions bridged by the U6 snRNA. Eluates were nuclease treated and eluted proteins were analyzed by data-independent acquisition mass spectrometry (Fig. 1d, e; and Supplementary Data 4). Consistent with previous results[45,46], THUMPD2 highly enriched its partner protein TRMT112, which is likely the only stable interactor (Fig. 1d–f). Among other proteins significantly enriched with THUMPD2, were various U6-interacting proteins, including known U6 snRNP biogenesis factors and early splicing factors (Fig. 1d, e). In line with the 3′ RACE data, none of the very early U6 biogenesis factors, such as La, TUT1 or USB1, were recovered.

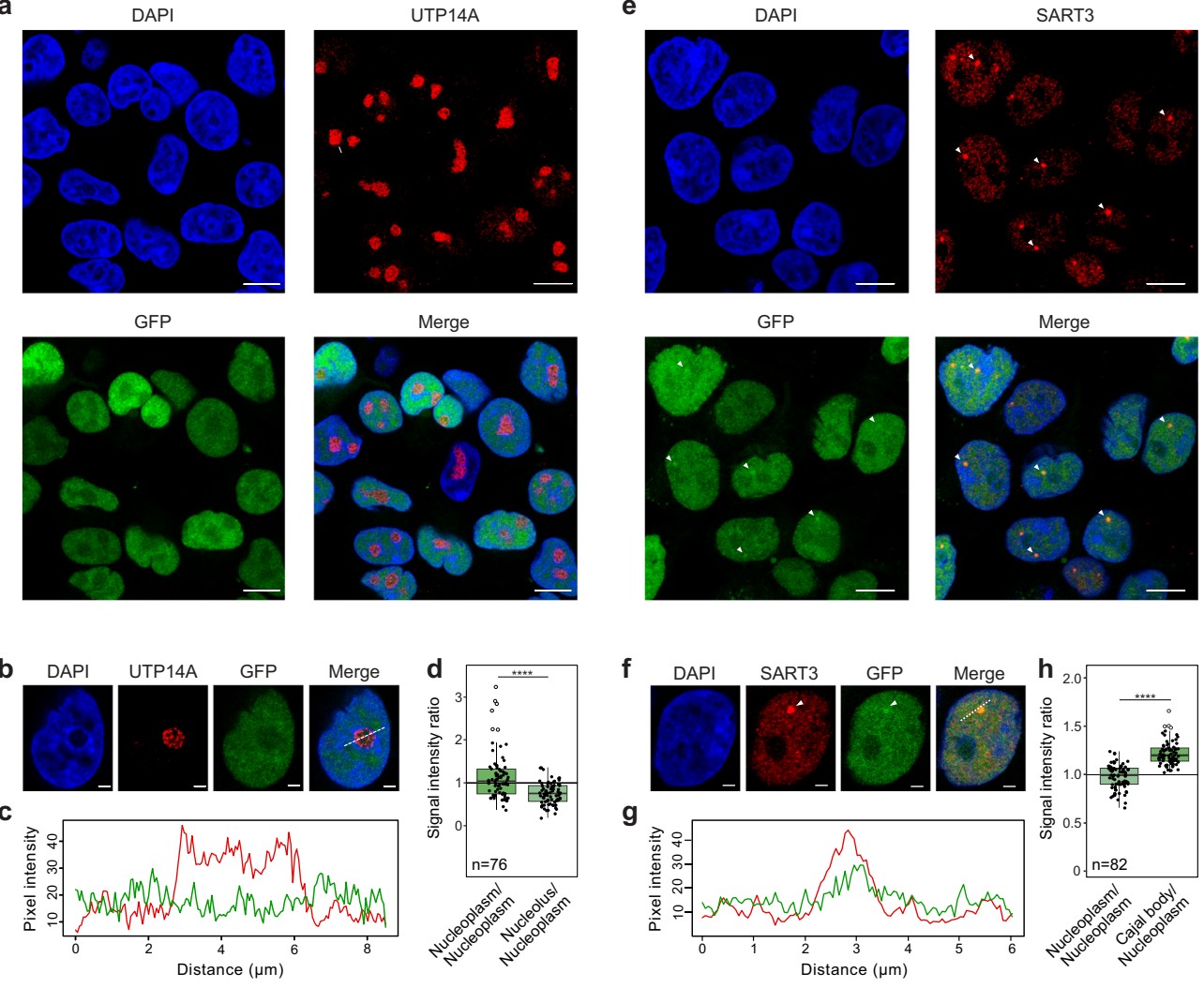

**Fig. 2 | THUMPD2 is enriched in Cajal bodies. a–h** HEK293 cells over-expressing GFP-THUMPD2 were used in immunofluorescence experiments (*n* = 3) where nucleoli (**a**) and Cajal bodies (**e**) were identified using UTP14A and SART3 anti-bodies, respectively, and nuclear material was stained with DAPI. White arrows in (**e**) indicate Cajal bodies. In (**b**) and (**f**), representative cells are shown indicating the cross-section used for the intensity plots in (**c**) and (**g**), respectively. Scale bars represent 10 μm (**a**, **e**) or 2 μm (**b**, **f**). In (**c**, **g**), intensity plots are shown representing the grey value distributions in the cross-section displayed in (b) and (f), respectively, compared to the distance in μm. Red lines correspond to the intensity in the red channel (UTP14A or SART3) and green lines corresponds to the intensity in the green channel (GFP). In (**d**) and (**h**), average grey values were quantified in nuclear bodies and the surrounding region (nucleoplasm) as in (**c**) and (**g**), respectively. UTP14A/SART3 staining was used to delimitate each nuclear body and mean grey values were obtained in the green channel (GFP) for the area within the body and the same area in the nucleoplasm. Number of cells used for quantification are indicated below the box plots and dots represent individual data points. Box limits represent the 25th to 75th percentiles, lines indicate the medians, whiskers the minimum and maximum and outliers are represented as white dots. p-values were calculated using unpaired two-tailed t-tests (****$p$ < 0.0001). Source data and $p$ values are provided as a Source Data file.

Focusing on known U6-interactors, proteins that were relatively more abundant in the THUMPD2-His$_6$-2xFLAG sample compared to the control include the capping enzyme (MEPCE) and proteins related to the 2′-*O*-methylation of U6, such as core snoRNP proteins (FBL, NOP56 and NOP58) and U6 2′-*O*-methylation bridging factors (LARP7 and TFIP11) (Fig. 1d, e). In addition, early splicing factors that are part of the di-snRNP (e.g., PRPF31) or tri-snRNP (e.g., EFTUD2, PRPF8, PRPF6, CD2BP2, SNRNP200, SNRNP40) were relatively more abundant in the THUMPD2-His$_6$-2xFLAG sample compared to His$_6$-2xFLAG (Fig. 1e). Notably, MEPCE and EFTUD2 were recovered better with THUMPD2-His$_6$-2xFLAG after crosslinking, suggesting these interactions are transient or labile (Fig. 1f). Taken together, these data indicate that THUMPD2 associates with the U6 snRNP during the later stages of its biogenesis.

## THUMPD2 is enriched in Cajal bodies

U6 snRNP biogenesis takes place in the nucleoplasm and nucleoli before final maturation steps and assembly into di- and tri-snRNP complexes in Cajal bodies. THUMPD2 localizes to nuclei[46,52], but to determine the sub-nuclear localization of THUMPD2, immuno-fluorescence with GFP-THUMPD2 was performed. UTP14A and SART3 were used as markers for nucleoli and Cajal bodies, respectively, and co-localization with GFP-THUMPD2 was monitored (Fig. 2). GFP-THUMPD2 was not excluded from nucleoli (Fig. 2a), but the GFP sig-nal intensity in the nucleolar regions was lower than that of the sur-rounding nucleoplasm (Fig. 2b–d). By contrast, co-localization of GFP-THUMPD2 with Cajal bodies was observed (Fig. 2e); although GFP-THUMPD2 was detected throughout the nucleoplasm, the signal intensity of GFP-THUMPD2 was higher in Cajal bodies (Fig. 2f–h).

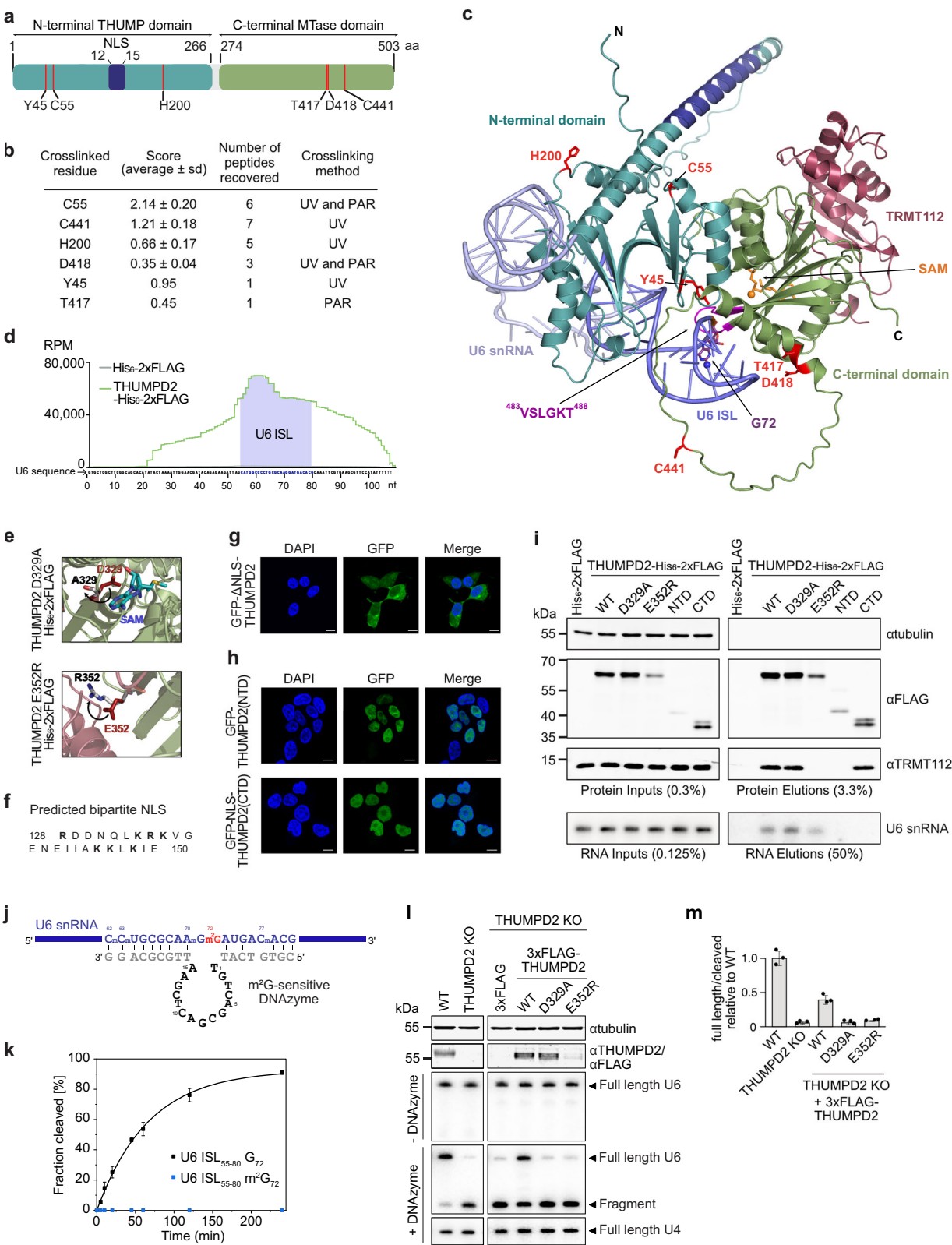

Taken together with the THUMPD2 interactome analyses (Fig. 1d–f) and the lack of effects of THUMPD2 depletion on U6 biogenesis steps taking place in nucleoli (Fig. 1b), the enrichment of THUMPD2 in Cajal bodies (Fig. 2e–h) suggests that the installation of m²G₇₂ could represent a final U6 snRNA maturation step before U6 snRNPs assemble into di-snRNPs in Cajal bodies.

## RNA contact sites on THUMPD2 are present in the THUMP and MTase domains

According to structure prediction of the THUMPD2-TRMT112 complex, the C-terminal Rossmann-fold methyltransferase (MTase) domain is accompanied by an N-terminal ferredoxin-like domain (NFLD)[53,54] and a THUMP fold composed of two α-helices packed

**Fig. 3 | The THUMP and MTase domains of THUMPD2 contribute to U6 association, and SAM-binding and TRMT112-interaction are necessary for m²G₇₂ installation. a** Scheme of THUMPD2 domain architecture (aa amino acid, MTase methyltransferase, NLS nuclear localization signal). Red lines indicate amino acids crosslinked to RNA in (b; confidence scores >0.15). **b** Cells over-expressing THUMPD2-His$_6$-2xFLAG were crosslinked (UV and PAR; $n = 1$ each) and protein-RNA crosslinked peptides in immunoprecipitations eluates were identified by mass spectrometry. **c** Model of U6 (light purple; $G_{72}$ – sticks, N2 atom – blue sphere) bound to the AlphaFold model of THUMPD2-TRMT112 (TRMT112 – raspberry; THUMPD2 - THUMP domain – cyan, MTase domain – light green, NLS – dark blue, $^{483}$VSLGKT$^{488}$ loop – magenta, SAM – orange (methyl group – sphere)). The crosslinked amino acids from (a, b) are identified in red. **d** Profile of CRAC reads mapping to each nucleotide of U6 is shown for one representative experiment. Normalized read counts are depicted as reads per million mapped reads (RPM). **e** Magnified view of amino acid substitutions anticipated to disrupt SAM binding (D329A) and TRMT112 interaction (E352R). **f** Predicted bipartite NLS in THUMPD2. **g, h** HEK293 cells over-expressing GFP-ΔNLS-THUMPD2 (**g**) or truncated versions of THUMPD2 N-terminally tagged with GFP (**h**) were used in fluorescence microscopy n = 3

experiments. Nuclear material was stained using DAPI (scale bars = 10 μm). **i** Lysates from cells over-expressing His$_6$-2xFLAG, THUMPD2-His$_6$-2xFLAG WT and variants (NTD and CTD) were used in $n = 4$ immunoprecipitation experiments. Proteins and RNAs in input and eluate samples were analyzed by western and northern blotting, respectively. **j** Schematic view of base-pairing of the m²G-sensitive DNAzyme with U6 snRNA to detect methylation of $G_{72}$. **k** U6 ISL RNA oligonucleotides (nucleotides 55–80) with $G_{72}$ or m²$G_{72}$ were incubated with the DNAzyme and the fraction of cleaved RNA at the indicated times was determined. Data from $n = 3$ independent experiments are shown as mean ± standard deviation. **l** Proteins from WT and THUMPD2 KO cells, and THUMPD2 KO cells expressing the indicated constructs were analyzed by western blotting. Tubulin served as a loading control, and antibodies against endogenous THUMPD2 and the FLAG tag were used in the left and right panels, respectively. Total RNAs were left untreated (-DNAzyme) or subjected to DNAzyme-mediated cleavage (+ DNAzyme). U6 and U4 snRNAs were detected by northern blotting. **m** Quantification of full-length U6 relative to the cleaved fragment after DNAzyme treatment. Ratios were normalized to WT samples. Data from $n = 3$ independent experiments (**l**) shown as mean ± standard deviation. Source data are provided as a Source Data file.

against a three-stranded β-sheet[55]. As the NFLD and THUMP fold are commonly found together and form a globular module, they are collectively termed the THUMP domain[56] (Fig. 3a). To better understand how the features of THUMPD2 contribute to interaction with the U6 snRNA, protein–RNA contact sites in THUMPD2 were mapped. Cells over-expressing THUMPD2-His$_6$-2xFLAG were crosslinked and after stringent enrichment of THUMPD2-His$_6$-2xFLAG, RNAs and proteins were digested, and THUMPD2 peptides crosslinked to nucleotides were identified by MS (Fig. 3a-b; and Supplementary Data 5). RNA contact sites were detected in both the THUMP and MTase domains (Fig. 3a-b).

To gain more insight into the role of these amino acids in U6 binding, a model of the THUMPD2-TRMT112 complex bound to a fragment of U6 was generated using AlphaFold[57]. In all the resulting models, the distance between U6-$G_{72}$ and the THUMPD2 active site was longer than 30 Å, meaning that these models are unlikely to be biologically significant. We therefore manually modeled U6 snRNA bound to THUMPD2-TRMT112 (Fig. 3c). This rigid body model is based on superposition of existing structures and is not biased by the experimentally identified THUMPD2–RNA crosslinking sites. In this model, no major steric clashes are observed, and the U6 region identified by CRAC as the THUMPD2 crosslinking site ($^{55}$CAUGGCmCCmCmUGCGC$^{58}$; Fig. 3d) is largely contacting amino acids from THUMPD2. Furthermore, U6-$G_{72}$ is nearby the SAM binding site and could easily project into the THUMPD2 active site by flipping out from the U6 ISL, as often seen in complexes containing double-stranded RNAs and RNA modification enzymes[59]. The highly conserved $^{483}$VSLGKT$^{488}$ loop of THUMPD2 is ideally positioned to disrupt the Watson-Crick base pair formed by $G_{72}$ with $C_{62}$ and project $G_{72}$ in the putative THUMPD2 active site.

Mapping the identified RNA contact sites onto the tertiary structure model revealed that, with exception of C55 and C441, the crosslinked amino acids are in close proximity to the RNA, suggesting a binding pocket for the U6 snRNA (Fig. 3c). However, C441 is located within a low complexity region proposed to adopt an extended conformation and hence could contact U6 snRNA bound to the THUMPD2-TRMT112 complex. Similarly, C55 is located in the vicinity of the 5′ and 3′ ends of the modeled U6 fragment and therefore, could interact with the extended U6 regions absent from the model.

### Both the THUMP and methyltransferase domains of THUMPD2 contribute to U6 interaction

To further explore the importance of different THUMPD2 features for binding and methylation of U6 as well as interaction with TRMT112, cell lines were generated for expression of THUMPD2 variants with amino acids substitutions within the *S*-adenosylmethionine (SAM)-binding motif (D329A) or the THUMPD2–TRMT112 interface (E352R) (Fig. 3e). In order to independently express the N- and C-terminal domains within the nucleus, the THUMPD2 region required for nuclear import was identified. A bipartite nuclear localization sequence (NLS) in the N-terminal domain of THUMPD2 was predicted[60] (Fig. 3f) and upon deletion of this sequence, GFP-THUMPD2-Δ128-150 remained in the cytoplasm (Fig. 3g). Truncated versions of THUMPD2 lacking the C-terminal MTase domain (NTD) or the N-terminal THUMP domain but with the NLS re-added (CTD) were therefore produced (Fig. 3h).

Next, THUMPD2-His$_6$-2xFLAG and its variants were used in IPs to monitor interactions with TRMT112 and the U6 snRNA. As expected, wild-type THUMPD2 recovered both TRMT112 and the U6 snRNA (Fig. 3i). The low amount of U6 snRNA in the IP eluate compared to the input likely reflects the long half-life of U6[61] and the transient interaction of THUMPD2 with nascent U6. The putative catalytically inactive mutant (D329A) interacted equally well as the wild-type protein with TRMT112, but recovered slightly more U6 (Fig. 3i), suggesting that perturbation of SAM binding and/or methylation by THUMPD2 may prolong its association with U6. E352R substitution abolished interaction between the MTase and cofactor, and reduced THUMPD2 expression (Fig. 3i), consistent with the described function of TRMT112 in shielding hydrophobic patches in associated methyltransferases to prevent their aggregation and degradation[62]. Interestingly, THUMPD2 E352R retained the ability to bind U6 (Fig. 3i), demonstrating that TRMT112 does not contribute directly to THUMPD2 association with its substrate. Consistent with the structural model, THUMPD2 NTD did not co-precipitate TRMT112 whereas THUMPD2 CTD associated with TRMT112, similarly to the full-length protein (Fig. 3i). Neither of the truncated forms of THUMPD2 were able to bind U6 (Fig. 3i), suggesting that the RNA contact sites in the N- and C-terminal regions (Fig. 3a-c) function cooperatively to enable stable association of the enzyme with its substrate.

### THUMPD2 SAM-binding and TRMT112-interaction are necessary for U6-m²$G_{72}$ installation

Demonstrating that the catalytic activity of THUMPD2 and its interaction with TRMT112 are necessary for formation of U6-m²$G_{72}$ in cells requires quantitative, site-specific detection of m²G in endogenous RNAs. Thus far, MS has been used for the detection of m²$G_{72}$ in isolated U6 snRNA (Fig. 1b)[45,46,63]. Deoxyribozymes (DNAzymes) are, however, emerging as sensitive tools allowing site-specific detection of modified nucleotides within total RNA[64]. In the course of developing methylation-sensitive DNAzymes by DZ-seq[65], we found that a variant of the 8-17 DNAzyme family[66] cleaves RNA in an m²G-sensitive manner (Fig. 3j). RNA oligonucleotides with the sequence of the U6 ISL, containing either $G_{72}$ or m²$G_{72}$, were used for DNAzyme-mediated

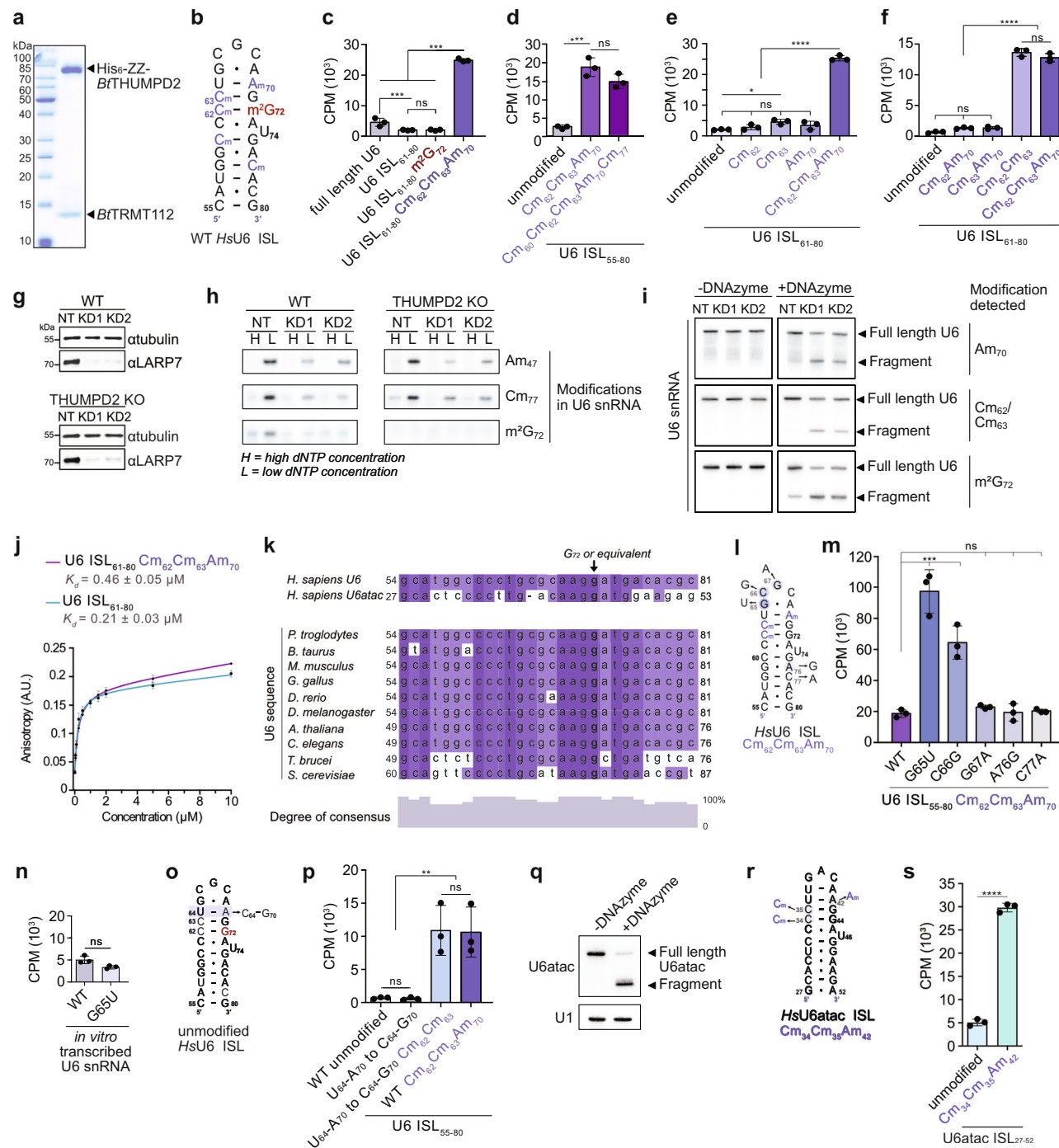

cleavage reactions. The $G_{72}$-containing oligonucleotide was efficiently cleaved by the DNAzyme, whereas the presence of $m^2G_{72}$ strongly impaired cleavage (Fig. 3k; and Supplementary Fig. S2a). Total RNA extracted from wild-type and THUMPD2 KO cells (Fig. 3l, left panels) was then incubated with the DNAzyme in cleavage buffer or left untreated, and U6 was monitored by northern blotting. In the absence of DNAzyme, only full length U6 snRNA was detected. Treatment of wild-type RNA with the DNAzyme lead to detectable amounts of a cleavage fragment, suggesting that $G_{72}$ is not fully methylated in cells (Fig. 3l, lower panels, left). However, U6 snRNA from the THUMPD2 KO cells was efficiently cleaved (Fig. 3l, lower panels, left), consistent with the LC-HRMS data (Fig. 1b)[45] and the sensitivity of the DNAzyme to $m^2G$.

To demonstrate the requirement of THUMPD2 SAM-binding and TRMT112 interaction for U6-$G_{72}$ methylation in cells, THUMPD2 KO

cells transiently expressing N-terminally 3xFLAG-tagged THUMPD2 variants (WT, D329A or E352R) or the 3xFLAG tag were prepared (Fig. 3l, upper panel, right). RNAs from the THUMPD2 KO cells expressing only the 3xFLAG tag showed U6 cleavage by the $m^2G$-sensitive DNAzyme equivalent to that observed for the untransfected THUMPD2 KO cells (Fig. 3l). Expression of wild-type 3xFLAG-THUMPD2 in the THUMPD2 KO cells restored the ratio of full-length:cleaved U6 to approximately 50% of that observed for wild-type cells (Fig. 3l-m), indicating partially rescued methylation of U6-$G_{72}$. However, when either THUMPD2 D329A or E352R was expressed, U6 was largely cleaved by the DNAzyme (Fig. 3l-m). Although the expression of THUMPD2 E352R was lower than wild-type, these data indicate that the methylation of $G_{72}$ requires the catalytic activity of THUMPD2 and that THUMPD2 interaction with TRMT112 is necessary for efficient methylation.

**Fig. 4 | 2′-O-methylations in the U6 ISL promote THUMPD2-TRMT112-mediated methylation of G$_{72}$. a** Co-purified His$_6$-ZZ-$Bt$THUMPD2 and $Bt$TRMT112 proteins separated by SDS-PAGE and visualized by Coomassie staining ($n = 3$). **b** Schematic view of the internal stem loop of human ($Hs$)U6 snRNA represented as free U6 with a bulged U$_{74}$ and C$_{61}$-A$_{73}$ wobble pair. m$^2$G$_{72}$ – red, 2′-O-methylated nucleotides (Nm) - purple. **c–f** In vitro methylation assays were performed using purified His$_6$-ZZ-$Bt$THUMPD2-$Bt$TRMT112 protein complex, [$^3$H]-SAM and the RNA substrates indicated in panels (**c**), (**d**), (**e**) and (**f**). Tritium incorporation was quantified and bar plots show averaged counts per minute (CPM) of $n = 3$ independent experiments with error bars representing mean ± standard deviation. Data were analyzed using one-way ANOVA and significance calculated using Tukey's multiple comparisons test. *$p < 0.05$, ***$p < 0.001$, ****$p < 0.0001$ and ns = not significant. **g** WT and THUMPD2 KO cells were transfected with a non-target siRNA (NT) or siRNAs against LARP7 (KD1 and KD2) and depletion of LARP7 was analyzed by western blotting ($n = 3$). **h, i** 2′-O-methylations and m$^2$G$_{72}$ in U6 snRNA after LARP7 knockdown were analyzed by primer extension (**h**; $n = 3$) or DNAzyme-catalyzed cleavage (**i**; $n = 3$) of RNA extracted from WT and THUMPD2 KO cells. **j** Fluorescence anisotropy measurements performed with purified His$_6$-ZZ-$Bt$THUMPD2-$Bt$TRMT112 and fluorescently-labelled unmodified and 2′-O-methylated (Cm$_{62}$, Cm$_{63}$, Am$_{70}$) U6 ISL

RNA oligonucleotides. Data from $n = 3$ independent experiments are shown as mean ± standard deviation and dissociation constants ($K_d$s) are given. (**k**) Sequence alignment of the U6 ISLs from different model organisms. Numbers indicate nucleotide position in the U6 sequence and colors indicate percentage of identity. **l** Scheme of the U6 ISL used for methylation assays in (**m**). 2′-O-methylated nucleotides–blue, nucleotide substitutions–arrows. **m** In vitro methylation assays performed as in (**c–f**) with the 2′-O-methylated wild-type U6 ISL or ISLs containing nucleotide substitutions in (**l**). Data were analyzed as in (**c–f**). **n** In vitro methylation assays performed as in (**c–f**) with in vitro transcribed wild-type or G$_{65}$U U6 snRNA. Results are presented as in (**c–f**). **o** Scheme of U6 ISL versions used for methylation assays in (**p**). **p** In vitro methylation assays performed as in (**c–f**) with U6 ISL depicted in (**o**). Data were analyzed as in (**c–f**). (**q**) Total RNA from HCT116 WT cells were untreated (-DNAzyme) or treated with an m$^2$G-sensitive DNAzyme targeting G$_{44}$ of U6atac ( + DNAzyme) ($n = 3$). U6atac and U1 were detected by northern blotting. **r** Scheme of the U6atac ISL used for methylation assays in (**s**). 2′-O-methylated nucleotides equivalent to Cm$_{62}$, Cm$_{63}$ and Am$_{70}$ in U6 are highlighted. **s** In vitro methylation assays performed as in (c-f) with the unmodified or 2′-O-methylated U6atac ISL shown in (**r**). Source data and $p$ values are provided as a Source Data file.

## 2′-O-methylated nucleotides in the U6 ISL promote THUMPD2-mediated modification of G$_{72}$

We next explored what elements in U6 endow specificity for THUMPD2-mediated modification. As human THUMPD2-TRMT112 expressed poorly in *E. coli*[45], recombinant His$_6$-ZZ tagged bovine ($Bt$) THUMPD2 and untagged TRMT112 were co-expressed and the heterodimeric complex was purified (Fig. 4a). The recombinant protein complex was incubated with in vitro transcribed U6 snRNA or chemically synthesized U6 internal stem-loop (ISL) fragments (nucleotides 61-80; Fig. 4b) together with [$^3$H]-SAM as a methyl group donor. Little methylation activity was detected towards the full-length U6 snRNA transcript, but the counts surpassed those obtained for a chemically synthesized U6 ISL containing m$^2$G$_{72}$ (Fig. 4c).

A notable feature of the U6 snRNA ISL is that it contains several 2′-O-methylated nucleotides (Fig. 4b). The enrichment of THUMPD2 in Cajal bodies (Fig. 2) suggests that THUMPD2-mediated methylation of U6 takes place after the 2′-O-methylation events. The methylation activity of THUMPD2 towards an unmodified ISL and an ISL containing the three 2′-O-methylations present in the endogenous 61-80 U6 ISL (Cm$_{62}$, Cm$_{63}$ and Am$_{70}$) was monitored, revealing that the 2′-O-methylated RNA was >10-fold more efficiently methylated by THUMPD2-TRMT112 in vitro (Fig. 4c). Further in vitro methylation assays using the complete ISL (55-80) and including the additional 2′-O-methylations (Cm$_{60}$ and Cm$_{77}$) showed that nucleotides 55-60 of U6 as well as 2′-O-methylation of C$_{60}$ and C$_{77}$ are dispensable for THUMPD2-mediated methylation (Fig. 4d).

To determine whether m$^2$G$_{72}$ installation is stimulated by a particular proximal 2′-O-methylated nucleotide or a combination of 2′-O-methylated nucleotides, ISLs with individual or pairs of 2′-O-methylated nucleotides were tested as THUMPD2-TRMT112 substrates. None of the individual 2′-O-methylated nucleotides substantially stimulated methylation by THUMPD2 (Fig. 4e). However, the substrate containing Cm$_{62}$ and Cm$_{63}$ was m$^2$G methylated equally well as the substrate containing three 2′-O-methylations, indicating that the combined presence of these two 2′-O-methylations on the G$_{72}$-opposing strand of the U6 ISL are sufficient to promote m$^2$G$_{72}$ formation by THUMPD2 (Fig. 4f).

As 2′-O-methylations within the ISL promote methylation by THUMPD2-TRMT112 in vitro, we explored whether this modification interdependence is also observed in cells. LARP7 is required for stoichiometric 2′-O-methylation of the U6 snRNA[67,68], so RNAi-mediated depletion was established in wild-type and THUMPD2 KO cells (LARP7 KD; Fig. 4g). Primer extension assays confirmed reduced 2′-O-methylation of various sites, including Am$_{47}$ and Cm$_{77}$, upon depletion of LARP7 (Fig. 4h; and Supplementary Fig. S3a-e). The m$^2$G$_{72}$-proximal 2′-

O-methylations Am$_{70}$, Cm$_{62}$ and Cm$_{63}$ could not reliably be detected, so 2′-O-methylation-sensitive DNAzymes targeting these sites were developed (Supplementary Fig. S2b-c). DNAzyme-mediated cleavage assays confirmed reductions in these 2′-O-methylations upon LARP7 KD (Fig. 4i). Importantly, supporting the in vitro methylation data (Fig. 4c-f), both primer extension and DNAzyme-mediated cleavage using the m$^2$G-sensitive DNAzyme demonstrated reduction of m$^2$G$_{72}$ in cells lacking LARP7 (Fig. 4h-i).

To investigate whether the enhanced methylation of the 2′-O-methylated U6 ISL arises due to a more stable interaction of THUMPD2 with the modified substrate, the affinity of THUMPD2-TRMT112 for U6 ISL RNA oligonucleotides unmodified or containing Cm$_{62}$, Cm$_{63}$ and Am$_{70}$ was determined by fluorescence anisotropy (Fig. 4j). The dissociation constants ($K_d$) were similar for both oligonucleotides (Fig. 4j), indicating that THUMPD2-TRMT112 binds both RNAs well and that the 2′-O-methylated nucleotides in U6 affect THUMPD2 methylation activity rather than substrate binding.

## Stabilization of U6 ISL stem does not bypass the requirement of Cm$_{62}$ and Cm$_{63}$ for m$^2$G$_{72}$ installation

It was reported that methylation of U6 requires a specific sequence in the apical loop of the ISL and a bulged structure close to G$_{72}$ formed by protrusion of U$_{74}$[46]. Comparing the sequence of the human ($Hs$)U6 ISL with that of $Hs$U6atac and other eukaryotes highlighted regions of lower conservation in the ISLs of *T. brucei*, *S. cerevisiae* and $Hs$U6atac, the latter two known to lack an m$^2$G modification[46] (Fig. 4k). To investigate the relevance of these sequences in U6 ISLs closely resembling the endogenous THUMPD2 substrate, in vitro methylation assays were performed with 2′-O-methylated U6 ISL sequences containing single-nucleotide substitutions in the apical loop and the stem (Fig. 4l). In contrast to the previous findings for the unmodified U6 snRNA transcript[46], in the context of the 2′-O-methylated U6 ISL, none of the nucleotide substitutions impeded methylation by THUMPD2 (Fig. 4m). Instead, the G$_{65}$U and C$_{66}$G substitutions in the apical loop increased THUMPD2-mediated methylation compared to the wild-type 2′-O-methylated ISL, while the nucleotide substitutions in the stem (A$_{76}$G and C$_{77}$A) did not affect m$^2$G$_{72}$ methylation (Fig. 4m). Importantly, we demonstrated that an unmodified transcript containing the G$_{65}$U substitution was not methylated by THUMPD2 more efficiently than the wild-type transcript (Fig. 4n).

Ribose 2′-O-methylations generally have a stabilizing effect on RNA structures[58], and in the context of a stem containing Cm$_{62}$, Cm$_{63}$ and Am$_{70}$, the G$_{65}$U and C$_{66}$G substitutions that extend base-pairing of the stem enhance THUMPD2-mediated methylation of G$_{72}$. To explore how stem stabilization influences THUMPD2 activity, in vitro

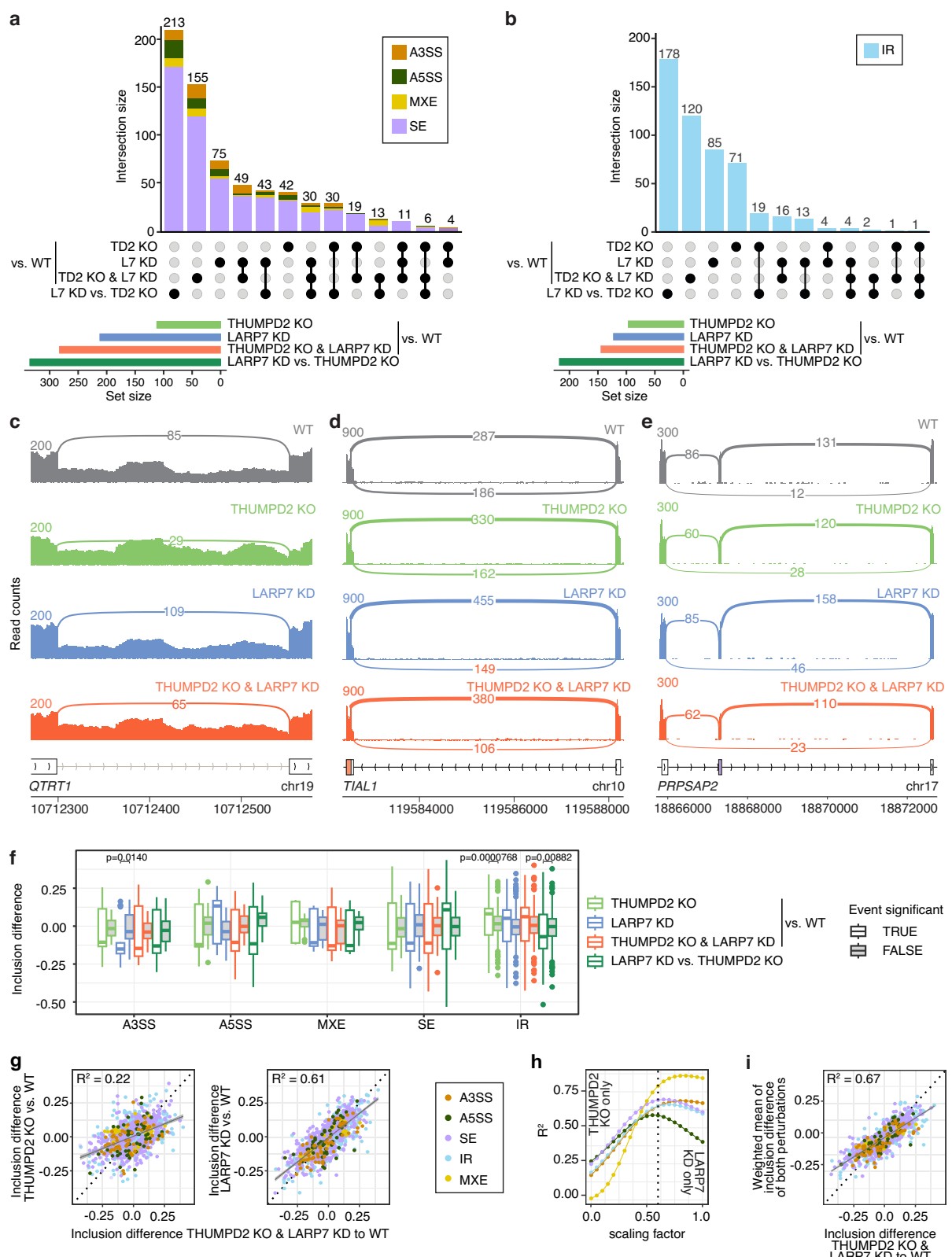

methylation assays were performed with a U6 ISL where the $U_{64}$-$A_{70}$ pair was mutated to a $C_{64}$-$G_{70}$ pair (Fig. 4o-p). When the 2′-O-methylations were lacking, this substrate was not methylated by THUMPD2-TRMT112, but when $Cm_{62}$ and $Cm_{63}$ were included, methylation levels of $G_{72}$ were similar to the WT 2′-O-methylated stem (Fig. 4p). Further strengthening of the stem by numerous nucleotide substitutions ($C_{61}U$, $G_{65}U$, $C_{66}G$ and $A_{76}C$; Supplementary Fig. S3f-g) also did not

promote $m^2G$ formation in vitro, emphasizing the specific importance of 2′-O-methylation of $Cm_{62}$ and $Cm_{63}$ as a recognition feature for THUMPD2-mediated methylation of U6-$G_{72}$.

In contrast to U6, U6atac does not contain 2′-O-methylations or an $m^2G$ within the ISL[46,69]. Consistent with the lack of $G_{44}$ (equivalent to U6-$G_{72}$) methylation in cells, which was confirmed using an $m^2G$-sensitive DNAzyme targeting U6atac (Fig. 4q), the unmodified

**Fig. 5 | THUMPD2 and LARP7 independently and interdependently affect alternative splicing.** RNAs from HCT116 WT and THUMPD2 KO cells treated with a non-target siRNA or one targeting LARP7 (LARP7 KD) were subjected to mRNA-seq (*n* = 3 independent experiments) and alternative splicing analysis was performed. **a, b** UpSet plots showing the number of alternative splicing events detected as significant in the indicated comparisons with respect to wild-type (WT) or between LARP7 KD (L7 KD) and THUMPD2 KO (TD2 KO). **a** Significant AS events quantified with rMATS (FDR < 0.01, absolute inclusion difference > 0.1, median read count in at least one condition > 50). **b** Significant IR events based on splicing efficiency calculation of annotated introns (*p* value < 0.01, absolute inclusion difference > 0.05, median read count in at least one condition > 50). **c** IR event specifically regulated by THUMPD2. **d** A3SS event specifically regulated by LARP7. **e** Skipped exon event with more exon skipping in any condition than WT. **f** Distribution of significant and not significant inclusion differences for different AS types and conditions. Statistical analysis was performed using the two-sided Wilcoxon rank-sum test and Bonferroni correction. Box limits (hinges) represent the 25th to 75th percentiles (Interquartile Range (IQR)), lines indicate medians, whiskers extend from the hinge to the minima and maxima that are not further than 1.5x IQR from the hinge, and outliers are represented as dots. **g** Correlations between inclusion level differences between THUMPD2 KO (left) / LARP7 KD (right) respective to WT to inclusion level differences resulting from the combination of THUMPD2 KO & LARP7 KD in cells. Error band represents the 95% confidence interval (CI) for the fitted linear regression line. **h, i** Inclusion level differences resulting from the combination of THUMPD2 KO & LARP7 KD in cells is better explained by a weighted combination of the individual lack of either THUMPD2 or LARP7 in cells. A scaling factor of 0.6 (dashed line in (**i**)) improves the overall correlation the most. **i** Correlations between inclusion level differences resulting from the combined THUMPD2 KO & LARP7 KD. Error band represents the 95% confidence interval (CI) for the fitted linear regression line.

U6atac ISL was a poor substrate for in vitro methylation by THUMPD2-TRMT112 (Fig. 4r-s). Upon 2′-*O*-methylation of $C_{34}$, $C_{35}$ and $A_{42}$ (equivalent to $Cm_{62}$, $Cm_{63}$ and $Am_{70}$ of U6, respectively) (Fig. 4r), the U6atac ISL was, however, readily methylated by THUMPD2-TRMT112 (Fig. 4s), further supporting the importance of specifically placed 2′-*O*-methylations for THUMPD2-mediated methylation of U6-$G_{72}$.

## $m^2G_{72}$ and 2′-*O*-methylations in U6 independently and interdependently influence alternative splicing

As installation of $m^2G_{72}$ requires the prior presence of 2′-*O*-methylations in the ISL, we investigated whether these modified nucleotides display synergistic and/or complementary effects on the functional level. We used RNA-seq to compare the effects of lack of U6 2′-*O*-methylation (LARP7 KD) with the absence of $m^2G_{72}$ (THUMPD2 KO) on alternative splicing (Fig. 5). Based on differences in overall transcript levels, the combined lack of THUMPD2 and LARP7 yielded a more pronounced difference in overall gene expression than lack of either of these proteins individually (Supplementary Fig. S4a; Supplementary Data 6).

Alternative splicing (AS) analysis allowed detection of five different types of AS, namely skipped exons (SE), alternative 3′ or 5′ splice site usage (A3SS and A5SS), mutually exclusive exons (MXE) and intron retention (IR; Supplementary Fig. S4b). Considering four comparisons that allow to evaluate the impact of individual versus combined effects of THUMPD2 KO and LARP7 KD, numerous significant AS events were detected, ranging from 0.3-2.1% of all events with sufficient sequencing read coverage (Fig. 5a, b; and Supplementary Data 7, 8). The fewest events were detected for the THUMPD2 KO (0.3-0.7% of all quantified events), consistent with a modest effect on alternative splicing by removing a single RNA modification from the U6 snRNA. LARP7 KD, which affects multiple methylation sites in U6 snRNA (Fig. 4h, i; and Supplementary Fig. S3a–e), yields more significant events (Fig. 5a, b; 0.3–1.3% of all quantified events). The combined absence of LARP7 and THUMPD2 yielded even more significant events than either of the individual depletions compared to wild-type (0.4-1.8% of all quantified events). Examples of AS events significantly affected by THUMPD2 KO, LARP7 KD or both are shown in Fig. 5c–e. Interestingly, the direct comparison of LARP7 KD vs. THUMPD2 KO identified the largest group of significant AS events (Fig. 5a, b; 0.6-2.1% of all quantified events). More in-depth analysis of individual AS events illustrates that most of the events highlighted by the THUMPD2 KO vs. LARP7 KD comparison are also significant by false discovery rate (FDR) in the THUMPD2 KO or LARP7 KD, but do not pass the inclusion difference cutoff when comparing to wild-type (Supplementary Fig. S4c). Across the different conditions, splice site strength varies slightly with the degree of intron retention and significant differences in intron retention are detected in introns that exhibit naturally inefficient pre-mRNA splicing (Supplementary Fig. S4d, e). Furthermore, THUMPD2 KO specifically increases intron retention as reported previously[45] (Fig. 5c, f; and Supplementary

Fig. S4e; Supplementary Data 8) and LARP7 KD affects A3SS in a directed manner with usage of distal 3′ SSs becoming more pronounced (Fig. 5d, f). Overall, these subtle, but significant differences between THUMPD2 KO and LARP7 KD suggest functionally distinct consequences of U6 snRNA $m^2G_{72}$ and 2′-*O*-methylations on AS and additive effects when combined.

For additive effects arising from co-regulated, as well as independently affected sets of AS events, the combined effects from lack of both proteins might be explainable by the weighted sum of individual effects. Both inclusion level differences upon THUMPD2 KO and LARP7 KD, respectively, correlated positively with inclusion level differences upon THUMPD2 KO & LARP7 KD in combination (Fig. 5g), but with the correlation for LARP7 KD being substantially stronger. The combination of both individual inclusion level differences weighted by a factor of 0.4 for THUMPD2 KO and 0.6 for LARP7 KD that was derived from probing individual weighted sums for individual AS types increased the overall correlation (R) and coefficient of determination ($R^2$) relative to the individual comparisons (Fig. 5g–i). Interestingly, the coefficient of determination improves more for some types of AS than others (Fig. 5h, i); as SE and IR events form the largest group of significant AS events, these dominate the overall coefficient of determination.

Overall, both the absence of THUMPD2 and depletion of LARP7 affect alternative splicing with more introns and exons being differentially spliced in the combination of both perturbations, which is in part explainable by THUMPD2 KO increasing intron retention and LARP7 KD enhancing distal A3SS usage.

## $m^2G_{72}$ and 2′-*O*-methylations in the U6 ISL differentially influence snRNP assembly

To explore the basis of the splicing defects arising from lack of $m^2G$ modification and/or these 2′-*O*-methylations, we first investigated their influence on the stability and structure of the U6 ISL in vitro as alterations might ultimately affect snRNP assembly and thereby pre-mRNA splicing. The folding of the U6 ISL alone and the stability of the U4/U6 duplex were first studied by UV thermal melting analysis. For the U6 ISL, three synthetic RNAs (nucleotides 55-80) were compared: the unmodified RNA, the analogue that carries $Cm_{62}$, $Cm_{63}$ and $Am_{70}$, and the analogue that additionally contains $m^2G_{72}$ (Fig. 6a). The melting curves were very similar, with almost the same hyperchromicity and nearly superimposable heating and cooling ramps (Fig. 6b). A similar situation was observed for the partial duplex structure that mimics the U4/U6 di-snRNP. Melting curves could clearly distinguish the U6 ISL from the di-snRNP mimic, but in neither case were significant shifts to higher melting points observed for the modified *vs.* unmodified samples (Supplementary Fig. S5a, b). However, further analysis of U6 ISLs by $^1$H-NMR spectroscopy clearly demonstrated an influence of the modifications on the ISL structure (Fig. 6c, d). Both the 2′-*O*-methylated RNA and the 2′-*O*-methylated RNA also containing $m^2G_{72}$ showed more signals than the unmodified RNA (Fig. 6c),

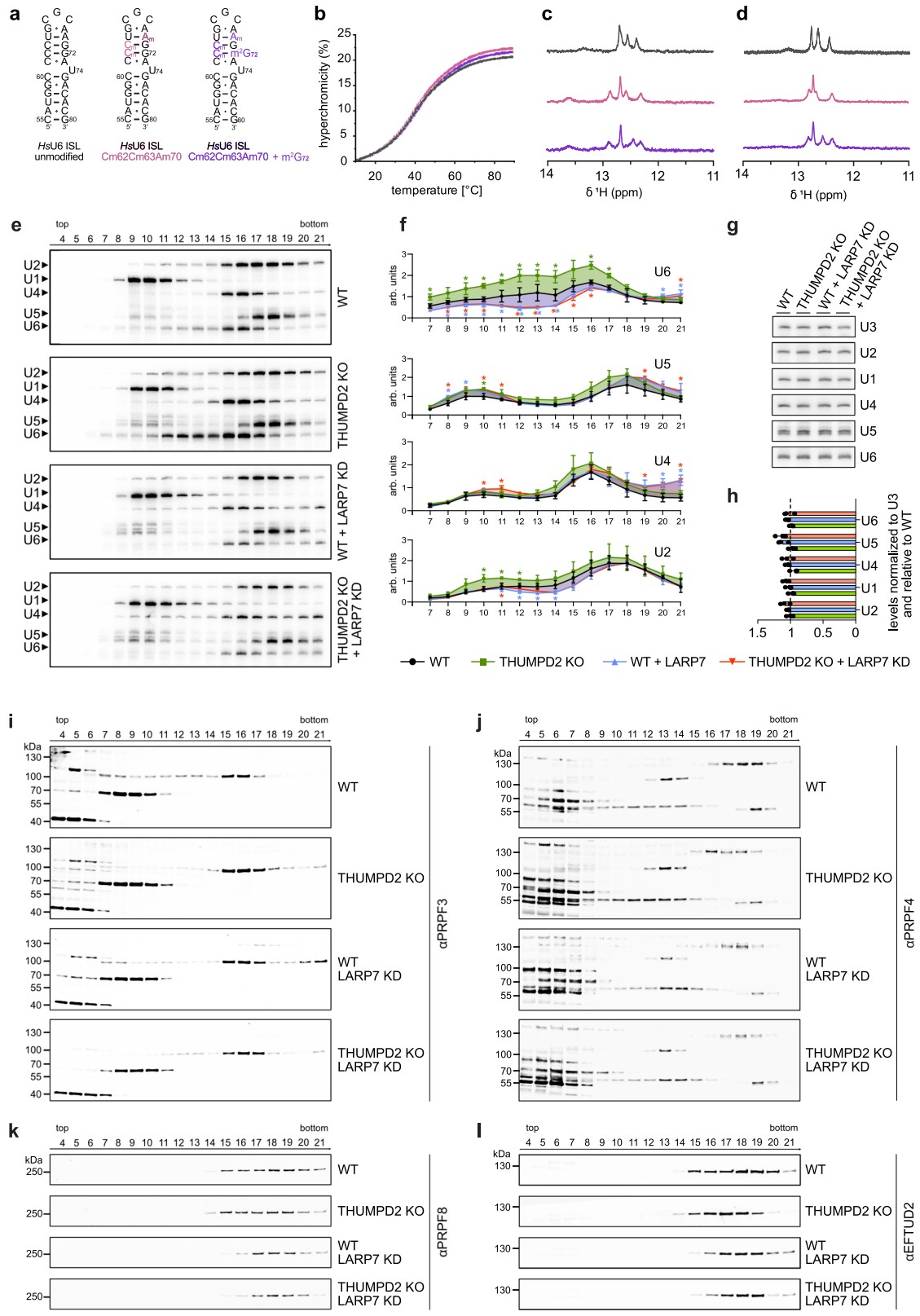

indicating that more nucleobase protons are protected from water exchange in the modified RNAs. The [1]H NMR spectra were recorded at 283, 298 and 308 K, in absence of $Mg^{2+}$ or with 1 or 2 mM $Mg^{2+}$ added (Supplementary Fig. S5c). The signal intensity generally decreased at higher temperatures due to enhanced base-pair fraying or partial melting of the hairpin structure. Indeed, barely any signals were detectable at 308 K in the absence of $Mg^{2+}$, while the addition of $Mg^{2+}$

clearly stabilized the RNA structures (Fig. 6d; Supplementary Fig. S5c). The effect was more pronounced for the modified RNAs than for the unmodified ISL, consistent with a stabilizing effect of the modifications. Although these results demonstrate an influence of these modifications on the U6 ISL structure, the contributions of ribose vs. nucleobase methylation could not be distinguished under the conditions tested.

**Fig. 6 | m²G₇₂ and 2′-O-methylations affect U6 ISL conformation and lack of these modifications impacts assembly of snRNP complexes. a** Schemes of synthetic RNAs used for biophysical investigations. Modified nucleotides are shown in pink/purple. **b** UV thermal melting curves plotting hyperchromicity at 260 nm. Conditions: 1 μM RNA, 100 mM NaCl, 10 mM potassium phosphate buffer pH 7.4, T = 10 °C – 90 °C. One of n = 4 reversible heating and cooling curves is shown. **c** ¹H-NMR spectra of imino region, 120 μM RNA in 10 mM potassium phosphate buffer pH 7.4, 90% $H_2O$/10% $D_2O$, 298 K. **d** Same as (**c**) but with 2 mM $MgCl_2$ and spectra recorded at 311 K. Number of signals correlates with number of base pairs. Color code according as in (**a**). **e, f, i–l** Sedimentation of snRNP particles from WT and THUMPD2 KO cells in glycerol gradients analyzed by northern (**e**) and western blotting (**i–l**). Numbers on top represent gradient fractions. **e** Northern blotting analysis of the levels of U1, U2, U4, U5 and U6 snRNAs (indicated by arrows). Representative image of n = 3 independent experiments. **f** Signal intensities of the snRNAs represented in (**e**) are shown for gradients performed with WT (black)

THUMPD2 KO (green), WT + LARP7 KD (blue) and THUMPD2 KO + LARP7 KD (red) cells as line plots. Shaded areas display fold changes from each sample relative to WT in their respective colors. Values were normalized to the average intensity of the U1 snRNA in each blot. Data from n = 3 independent experiments are shown as mean ± standard deviation. Statistical analysis was performed using multiple unpaired two-sided t-tests (*$p < 0.05$) Arb. Units – arbitrary units. **g** Northern blotting analysis of the U1, U2, U4, U5 and U6 snRNAs and U3 snoRNA (loading control) in the indicated nuclear extracts. Representative image of n = 3 independent experiments. **h** Quantification of the snRNA levels in (**g**) as bar plots where error bars represent mean ± standard deviation. Signal intensities were normalized to U3 and are shown relative to WT. **i–l** Western blotting analysis of PRPF3 (di-snRNP marker) (**i**), PRPF4 (di-snRNP marker) (**j**), PRPF8 (tri-snRNP marker) (**k**) and EFTUD2 (tri-snRNP marker) (**l**) are shown. Representative images of n = 3 experiments. Source data and p values are provided as a Source Data file.

As the methylations influence the structure of the U6 ISL, we addressed whether m²G₇₂ and/or the U6 2′-O-methylations affect the assembly of di-(U4/U6)/tri-(U4/U6•U5)snRNPs in cells. Nuclear extracts prepared from wild-type and THUMPD2 KO cells, depleted or not of LARP7 (LARP7 KD), were separated on glycerol gradients and the migration patterns of the snRNAs were determined by northern blotting (Fig. 6e, f). The di-snRNP proteins PRPF3 (Fig. 6i) and PRPF4 (Fig. 6j), which become SUMOylated and ubiquitinated during snRNP maturation[70,71], as well as the U5 snRNP-associated components of the tri-snRNP PRPF8 (Fig. 6k) and EFTUD2 (Fig. 6l) were also detected as additional markers of these complexes. It should be noted that due to the requirement of C₆₂/₆₃ 2′-O-methylation for efficient THUMPD2-TRMT112-mediated methylation (Fig. 4c–i), when LARP7 is depleted in the wild-type background, m²G₇₂ is strongly reduced.

In the THUMPD2 KO, the U6 snRNA moderately accumulated in fractions (f)10-17 (Fig. 6e, f, upper two panels of e) relative to the U1 snRNP, which was detected with a similar migration profile to wild-type. As the absence of THUMPD2 does not impact the overall level of U6 (Fig. 6g, h)[45], this suggests that an increased proportion of U6 snRNA is retained in emerging snRNP complexes when G₇₂ is not modified. The concomitant accumulations of U4 in di-snRNPs (f15-16) and U5 in tri-snRNPs (f17-19) in THUMPD2 KO cells indicate that lack of U6-m²G₇₂ affects assembly of the U6 snRNP into di-/tri-snRNP complexes and their progression into larger spliceosomal assemblies (e.g., B complex; not detectable on these gradients). Consistent with the notion of altered snRNP assembly in cells lacking THUMPD2, a marked decrease in the progression of unmodified PRPF3 to the most modified form in f9-14 was observed, as well a slight accumulation of this protein in late fractions (f20-21; Fig. 6i). The altered migration profile of PRPF4 with higher levels in f11-14 (leading to di-snRNPs) concomitant with reduced amounts reaching tri-snRNP complexes (f17-19) further supports the accumulation of intermediates on the pathway towards formation of the tri-snRNP in the absence of THUMPD2 (Fig. 6j). Interestingly, in THUMPD2 KO cells, a mild increase in the amount of U6atac was also observed in f12-16 (Supplementary Fig. S6a). As U6atac is not a substrate of THUMPD2 (Fig. 4q) and the overall level of U6atac was unaffected by loss of THUMPD2 (Supplementary Fig. S6b), it is possible that this effect arises due to the observed alterations in PRPF3/4 as these proteins associate with both U6 and U6atac snRNPs.

In contrast to THUMPD2, upon depletion of LARP7, a strong decrease in the amount of U6 present in f10-17 was observed (Fig. 6e-f, lower two panels of e). As LARP7 KD did not affect the overall level of U6 (Fig. 6g), this highlights the requirement of the U6 2′-O-methylations for snRNP assembly. The marked decrease in U6 incorporation into snRNPs was accompanied by alterations in the migration profiles of U4 and U5; an accumulation of both U4 and U5 was observed in f20-21 (Fig. 6e-f), potentially indicating impaired association of tri-snRNP complexes with pre-mRNAs. Similar to the U4 and U5 snRNAs, the modified form of PRPF3 was also accumulated in f18-21 upon LARP7

KD. Overall levels of PRPF4 in f17-19 were decreased in cells lacking LARP7 (Fig. 6j), and both PRPF8 and EFTUD2 were strongly decreased (f15-16), exhibiting a clear shift in the gradients (Fig. 6k, l).

The effects on snRNP assembly observed upon LARP7 depletion in the wild-type background were mirrored when LARP7 depletion was combined with THUMPD2 KO (Fig. 6e-l), consistent with loss of U6-m²G₇₂ in both contexts. Phenotypically, the lack of U6 2′-O-methylations caused by LARP7 KD appears to predominate over the absence of m²G in the THUMPD2 KO. Taken together, these results demonstrate that lack of m²G₇₂ and the 2′-O-methylations differently affect U6 incorporation into the snRNPs, but loss of either converges into less competent snRNPs assembled, negatively affecting pre-mRNA splicing.

## Discussion

In higher eukaryotes, the U6 ISL is densely decorated with modifications that include several 2′-O-methylations and an m²G, but mechanistic insights into how the U6 ISL modification landscape is established were lacking as well as knowledge on how these modified nucleotides impact spliceosome assembly and function. Several snRNPs installing U6 ISL 2′-O-methylations were already predicted and validated years ago[29,72–74], and now, RNA–RNA crosslinking analyses have highlighted possible snoRNA candidates guiding the remaining 2′-O-methylations (Supplementary Fig. S7a)[75,76]. THUMPD2 was also recently revealed as the MTase responsible for N2 methylation of G₇₂[45,46]. THUMPD2 shares a similar domain architecture to two other human m²G MTases (THUMPD3 and TRMT11), but while they each modify multiple tRNA targets, our data indicate THUMPD2 has specificity for the U6 snRNA. Structural analyses of THUMP domain-containing tRNA modification enzymes revealed that the THUMP domain interacts with the tRNA acceptor stem and the CCA tail, enabling it to direct modification of specific target nucleotides within the tRNA acceptor stem/D-arm by the associated methyltransferase/pseudouridine synthetase/thiolase domains[53,54,77,78]. In this way, the THUMP domain acts as both an RNA binding platform and a molecular ruler, coupling modification enzyme recruitment to target nucleotide specificity. In the prokaryotic THUMP domain-containing proteins Trm11, Trm14 and TrmN, the THUMP domain is a continuous entity[53] whereas in THUMPD2, a long insert is present in this domain. Mapping of RNA crosslinking sites on THUMPD2 highlighted H200 within this insert, suggesting that the THUMP domain of THUMPD2 may be specifically optimized for interaction with its U6 snRNA substrate. Although not experimentally derived, in the model of the THUMPD2–TRMT112–U6 ISL complex, the mode of interaction of the THUMPD2 THUMP domain with U6 shares parallels with tRNA binding by other THUMP domains; it appears to rely on recognition of structural features, such as double-stranded regions of the substrate RNA, rather than a sequence motif. Notably, neither the THUMP nor MTase domains of THUMPD2 stably associate with the U6 snRNA independently, indicating that they function synergistically in substrate binding. This is in line with the previous

observations that the THUMP domains of some $m^2G$ tRNA methyltransferases cannot or only very weakly bind RNA autonomously[79,80]. As perturbation of the THUMPD2 interaction with TRMT112 does not impede U6 interaction, this suggests that the cofactor stabilizes the methyltransferase and promotes catalysis, but does not substantially contribute to target recognition or substrate binding.

To date, U6-$m^2G_{72}$ has been detected using MS, which requires large amounts of purified RNAs to precisely map the modification, and here, we introduce an $m^2G$-sensitive DNAzyme as an efficient, sensitive and quantitative tool for the direct, site-specific detection of $m^2G$ in cellular RNAs. Both MS analyses and DNAzyme-mediated cleavage assays indicate a high stoichiometry of U6-$m^2G_{72}$, but nevertheless, a small fraction of U6 snRNA from wild-type cells is cleaved by the DNAzyme, indicating the presence of some unmodified $G_{72}$ in endogenous U6; it is possible that the minimal cleavage of cellular U6 snRNA reflects either targeting of unmodified precursor U6 or the existence of mature U6 transcripts lacking $m^2G_{72}$. With the $m^2G$-sensitive DNAzyme, the requirement of SAM binding by THUMPD2 for the installation of U6-$m^2G_{72}$ was demonstrated, confirming THUMPD2 as catalytically responsible for methylating this position. Although THUMPD2 that is unable to interact with TRMT112 was partially destabilized, the remaining protein did not promote U6-$G_{72}$ methylation. This implies that the association of THUMPD2 with TRMT112 not only enhances the solubility and stability of THUMPD2 by shielding the hydrophobic surface of the methyltransferase domain, as has been observed for other TRMT112-methyltransferase complexes[52,62], but also promotes catalytic activity. In the case of *S. cerevisiae* Trm11 and human HEMK2, association with TRMT112 is mandatory for SAM binding[81,82], raising the possibility that inefficient THUMPD2-SAM complex formation due to the disrupted association with TRMT112 impacts methylation activity.

Our localization and interactome data, as well as the discovery that THUMPD2-mediated methylation is enhanced by 2′-*O*-methylations in the U6 ISL installed in the nucleolus, suggest that methylation of $G_{72}$ is one of the final U6 snRNA maturation events. Although most U6 modification events happen in the nucleolus, it is conceivable that U6 methylation takes place within Cajal bodies, where pseudouridylation of $U_{40}$ guided by SCARNA23/3 (Supplementary Fig. S7a) is also thought to occur. The discovery that the stem portion of the ISL is required for efficient THUMPD2-mediated methylation indicates that, if U6-$G_{72}$ is methylated in Cajal bodies, this modification takes place prior to di-snRNP formation as the U6 ISL is unwound by binding of SART3 to allow formation of alternative base-pairing with U4.

The finding that THUMPD2 preferentially modifies a 2′-*O*-methylated substrate is analogous to other modification circuits, such as the $m^3C_{32}$ tRNA modifications that require the presence of $i^6A/t^6A_{37}$ for efficient installation[83,84], and the interdependent $m^5C_{38}$ and $Q_{34}$ modifications detected in tRNA$^{Asp}$[85,86]. Considering how the 2′-*O*-methylations in the U6 ISL stimulate $m^2G_{72}$ installation, it is notable that in the THUMPD2-TRMT112-U6 model, $Cm_{62}$ and $Cm_{63}$ are oriented towards the solvent. It is possible that the long flexible loop in the THUMPD2 C-terminal MTase domain, which contains C441 that was identified as an RNA crosslinking site, folds back onto this face of the U6 ISL, potentially allowing direct recognition of the 2′-*O*-methylated nucleotides by THUMPD2. However, as ribose 2′-*O*-methylations favor the C3′ endo ribose pucker, it is also possible that $Cm_{62}/Cm_{63}$ enhance THUMPD2-mediated methylation by influencing the structure of the U6 ISL. Although extending the stem region of a 2′-*O*-methylated ISL stimulates methylation by THUMPD2-TRMT112, nucleotide substitutions that stabilize the 2′-*O*-methylated stem are ineffectual. This indicates that increased stem stability does not per se promote THUMPD2-mediated methylation, but that methylation of the ribose moieties of $C_{62/63}$ in the strand opposite to $G_{72}$ endows a specific property that optimizes the ISL as a substrate. 2′-*O*-methylations can alter conformational preferences of flexible RNAs containing bulges and internal loops[87], so it is possible that persistence of a specific

conformational state of the ISL, favored by the presence of the 2′-*O*-methylations, renders the RNA a better substrate for THUMPD2. As the affinity of THUMPD2 for a non-modified U6 ISL and one containing 2′-*O*-methylations are comparable, this implies that RNA binding by THUMPD2 and its methylation activity are uncoupled, and the 2′-*O*-methylations within the U6 internal stem are important specifically for efficient methylation of $G_{72}$.

The evolutionary selection against a U6 ISL with a more stable stem is likely a result of the necessity for U6 to assume multiple conformations during its assembly and integration into spliceosomes. Mutational analysis of human U6 have shown that over-stabilization of the U6 ISL can be detrimental for U4-U6 interaction, snRNP assembly and splicing. For instance, exchange of the $C_{61}$-$A_{73}$ or $G_{59}$-$A_{76}$ pairs proximal to the bulged $U_{74}$ for C-G/G-C pairs, respectively, reduces U4-U6 interaction, and impairs spliceosome assembly and function[88,89]. In yeast, it has been shown that the presence of a rigid structure around the catalytic site can hinder formation of the catalytic triplex leading to a cold-sensitive phenotype, likely arising due to perturbed pre-mRNA splicing[90]. Compared to the well conserved upper region of the ISL, the lower ISL segment has a higher sequence variability, containing frequent weak and non-canonical base pairings that confer higher ISL flexibility[90]. The presence of 2′-*O*-methylations in the ISL region likely ensures an optimal balance between stability and flexibility that allows the required rearrangements of this sequence during the transitions between the different steps in snRNP assembly and splicing.

The lack of 2′-*O*-methylations in U6atac explains the specificity of THUMPD2 for the U6 snRNA[45]. The 2′-*O*-methylations in the U6 ISLs of higher eukaryotes are introduced by snoRNPs, in which the guiding snoRNA base-pairs with the target RNA. Notably, the regions of the U6 ISL that are poorly conserved between higher and lower eukaryotes contain the snoRNA base-pairing sites (Supplementary Fig. S7b). This raises the possibility of co-evolution of the snoRNA sequences and the sequence of the U6 ISL, as proposed previously for other snoRNAs and their targets[91]. In this context, the divergence of THUMPD2 from the other $m^2G$ methyltransferases could have occurred simultaneously making it a specialized U6 snRNA writer, sensitive to the 2′-*O*-methyl groups in the U6 ISL.

Analysis of alternative splicing in cells lacking $m^2G_{72}$ (THUMPD2 KO), 2′-*O*-methylations in U6 (LARP7 KD) or both indicates distinct roles of these modifications in modulating splicing patterns. The more pleiotropic effect of LARP7 KD compared to THUMPD2 KO is consistent with the lack of one *vs*. eight methylations, and likely also reflects the impact of 2′-*O*-methylations outside the ISL, which may influence interactions of the U6 snRNA with pre-mRNAs or spliceosomal proteins. The mild effects on alternative splicing, especially intron retention, observed in the absence of $m^2G_{72}$ are in line with the lack of a single methyl group. LARP7 KD increased the usage of distal 3′ splice sites, which could point to 2′-*O*-methylations providing an advantage for co-transcriptional spliceosome assembly on proximal 3′ splice sites[92]. The combined lack of both $m^2G_{72}$ and the 2′-*O*-methylations lead to the greatest number of alternative splicing events. This suggests that although these RNA modifications function independently, they also work additively to regulate pre-mRNA splicing.

The perturbation of splicing in the absence of THUMPD2 or LARP7 is rationalized by the detection of impaired snRNP assembly in cells lacking U6-$m^2G_{72}$ and/or the ISL 2′-*O*-methylations. Although the effects of $m^2G$ and 2′-*O*-methylations on U6 ISL structure and U4/U6 annealing are modest in vitro, marked differences could be detected in the dynamics of snRNP complexes assembly in cells. In the absence of THUMPD2, U6 accumulates in early snRNP assembly stages leading up to the formation of tri-snRNPs, implying that the presence of this modified nucleotide in the ISL facilitates formation of di-/tri-snRNP complexes, likely also affecting their integration into spliceosomal B complexes. Notably, differences are also observed in the distribution and/or modification status of PRPF3/PRPF4 di-snRNP and EFTUD2/

PRPF8 tri-snRNP proteins, supporting altered dynamics of snRNP complexes formation. It is tempting to speculate that installation of the $G_{72}$ methylation mark as a final milestone in U6 snRNP biogenesis may license the incorporation of U6 into di-/tri-snRNPs and spliceosomal complexes. As the accumulation of U6 snRNA in pre-spliceosomal snRNPs was more pronounced than that of the U4 and U5 snRNAs, it is also possible that increased incorporation of U6 snRNA into snRNPs represents an adaptation of the THUMPD2 KO cell line to compensate for the lack of U6-$m^2G_{72}$. The observed role for $m^2G_{72}$ in promoting efficient assembly of U6-containing snRNPs and their integration into later spliceosomal complexes is in line with the finding that overexpression of U6 partially rescues the defects in pre-mRNA splicing arising from lack of U6-$m^2G_{72}$[46]. Interestingly, the spliceosomal snRNPs assembly defects observed in the absence of U6-$m^2G_{72}$ are mirrored upon depletion of the G-patch protein TFIP11 from cells[93]. Knockdown of TFIP11, which reduced levels of a subset of U6 2'-O-methylations (including $Cm_{60}$ and $Cm_{62}$), leads to accumulation of the di-snRNP protein PRPF3 in pre-tri-snRNP fractions, an increase in the amount of free U5 snRNA and accumulation of tri-snRNP complexes, amongst other effects[93]. Strikingly, compared to the THUMPD2 KO, depletion of LARP7 differentially affects snRNP assembly, leading to a strong reduction in U6 incorporation into snRNPs. The finding that lack of $m^2G_{72}$ alone leads to accumulation of the U6 snRNA in early gradient fractions, whereas depletion of LARP7 strongly reduces incorporation of U6 into snRNPs suggests that the U6 2'-O-methylations are functionally important prior to $m^2G_{72}$ formation. Perturbed 2'-O-methylation of U6 is observed in Alazami syndrome patients carrying mutations in *LARP7*[68], so the finding that loss of these modifications impairs incorporation of U6 into snRNPs also contributes to understanding of the basis of this disease.

Taken together, our study provides important mechanistic insights into the hierarchical installation of RNA modifications in the spliceosomal catalytic core and also defines a role for $m^2G_{72}$ in ensuring efficient incorporation of U6 into spliceosomal complexes for optimal pre-mRNA splicing.

## Methods
### Molecular cloning
The codon optimized sequence of full-length *Hs*THUMPD2 and truncated versions were amplified from a pcDNA5-THUMPD2-His$_6$-2xFLAG vector[45] using oligonucleotides listed in Supplementary Data 9 and cloned by standard restriction digestion and ligation or Gibson assembly[94] into pcDNA5- or pCI-based vectors (Supplementary Data 10). The pcDNA5-based constructs were used to generate stably transfected HEK293 cell lines for the expression of C-terminally His$_6$-2xFLAG tagged or N-terminally GFP tagged THUMPD2 and variants. pCI-based plasmids were used for transient expression of THUMPD2 and variants with an N-terminal 3xFLAG tag in HCT116 cells. Site-directed mutagenesis using primers listed in Supplementary Data 9 was used to generate pcDNA5-based constructs for the expression of THUMPD2 with D329A and E352R amino acid substitutions. The mutant sequences were amplified using oligonucleotides in Supplementary Data 9 and cloned into pCI vectors (Supplementary Data 10). For the recombinant expression of *Bt*THUMPD2-TRMT112, a previously generated pET28-based construct was used[45]. Plasmids containing the U6 snRNA wild type and mutant sequences were generated by ligation of a recursive PCR product produced using four overlapping oligonucleotides (Supplementary Data 9). The pSP64-derived plasmids contain the sequence of the T7 promoter and directly downstream the sequence of U6 snRNA. All generated constructs were verified by Sanger sequencing (Microsynth).

### Human cell culture
HEK293 Flp-In™ T-REx™ (Thermo Fisher Scientific) were cultured in DMEM (Gibco) supplemented with 10% fetal bovine serum (Sigma-Aldrich) and 100 U/ml penicillin and 100 µg/ml streptomycin (Gibco) at 37 °C and 5% $CO_2$ and treated with blasticidin (10 µg/ml) every second passage. For the generation of stably transfected cell lines, HEK293 Flp-In T-REx cells were co-transfected with pcDNA5-based constructs (Supplementary Data 10) and the pOG44 Flp-recombinase expression plasmid (Thermo Fisher Scientific) using X-tremeGENE 9 DNA Transfection Reagent (Sigma-Aldrich) according to the manufacturer's instructions. Cells with correct transgene insertion into the Flp-In locus were selected with hygromycin (100 µg/ml) for two weeks. In established cell lines, expression of tagged proteins was induced by adding 1 µg/ml tetracycline 24 h before harvesting.

HCT116 cells were cultured in McCoy's 5 A medium (Gibco) supplemented with 10% fetal bovine serum (Sigma-Aldrich) and 100 U/ml penicillin and 100 µg/ml streptomycin (Gibco) at 37 °C and 5% $CO_2$. HCT116 parental and knockout cell lines used in this study were prepared and characterized as described before[45]. For transient transfection experiments, HCT116 cell lines were transfected with pCI mammalian expression vectors (Supplementary Data 10) using Lipofectamine 2000 (Thermo Fisher Scientific) according to the manufacturer's guidelines. Cells were passaged 24 h after transfection and selected for neomycin resistance using 800 µg/ml G418 (Sigma-Aldrich) for 5 days before harvesting.

### UV crosslinking and analysis of cDNA (CRAC)
For CRAC experiments[49,50], HEK293 cells over-expressing THUMPD2-His$_6$-2xFLAG or the His$_6$-2xFLAG tag were crosslinked using ultraviolet (UV) light at 254 nm (800 mJ/cm$^2$) using a Stratalinker (Stratagene). Cells were harvested and lysed by sonication in TMN150 buffer (50 mM Tris-HCl pH 7.8, 150 mM NaCl, 0.1% NP-40 (v/v), 5 mM β-mercaptoethanol supplemented with 1x cOmplete protease inhibitor cocktail (Roche). Protein-RNA complexes were affinity purified using Anti-FLAG M2 magnetic beads (Sigma-Aldrich) as described for the immunoprecipitation experiments and eluted overnight using 150 µg/ml 3x FLAG peptide (Sigma-Aldrich) in TMN150 buffer. The eluates were treated with 0.1 U RNace-IT Ribonuclease Cocktail (Agilent) for 30 sec at 37 °C to partially digest the RNAs. Complexes were immobilized on Ni-NTA beads (Qiagen) pre-equilibrated in wash buffer 1 (50 mM Tris–HCl pH 7.8, 300 mM NaCl, 0.1% (v/v) NP-40, 6 M guanidine hydrochloride, 10 mM imidazole and 5 mM β-mercaptoethanol). Complexes were incubated with beads under denaturing conditions for 2 h at 4 °C and washed with wash buffer 1 and PNK buffer (50 mM Tris–HCl pH 7.8, 10 mM $MgCl_2$, 0.5% (v/v) NP-40 and 5 mM β-mercaptoethanol). Co-purified RNA fragments were dephosphorylated using TSAP (Promega) for 30 min at 37 °C before on-bead ligation of the TruSeq Illumina RA3 3' adaptor (5'-/5rApp/TGGAATTCTCGGGTGCCAAGG /3ddC/-3') to the 3' end of the RNA overnight at 16 °C. The 5' end of the RNA fragments were labelled with [$^{32}$P] and the 5' adaptor (5'-/InvddT/ACACrGrUrUrCrArGrArGrUrUrCrUrArCrArGrUrCrCrGrArCrGrArUrCrNrNrNrNrNrArGrC-3; NNNNN is a unique molecular identifier or UMI). The crosslinked protein-RNA complexes were eluted in a buffer containing 50 mM Tris-HCl pH 7.8, 50 mM NaCl, 150 mM imidazole pH 8, 0.1% (v/v) NP-40 and 5 mM β-mercaptoethanol. The eluates were precipitated using trichloroacetic acid (TCA) and protein-RNA complexes were separated on a 4-12% NuPAGE Bis-Tris gel (Thermo Fisher Scientific) using MES SDS running buffer and transferred to a HyBond-C Extra nitrocellulose membrane (Amersham Biosciences) that was exposed to an X-ray film. After autoradiography, appropriate areas of the membrane were excised and RNA fragments were isolated by Proteinase K (Roche) treatment followed by phenol:chloroform:isoamyl-alcohol (25:24:1, v/v) extraction and ethanol precipitation. The recovered RNA was reverse transcribed with SuperScript III Reverse Transcriptase (Thermo Fisher Scientific) using the RTP primer (Supplementary Data 9). The resulting cDNAs were amplified using Phusion Hot Start II DNA-Polymerase (Thermo Fisher Scientific) and purified using AmpureXP beads

(Beckman Coulter). The cDNA libraries were separated by agarose gel electrophoresis and purified using the MinElute Gel Extraction Kit (Qiagen) according to manufacturer's instructions. The concentration was determined using a Qubit fluorometer (Thermo Fisher Scientific). Next-generation sequencing was performed at the NGS Integrative Genomics Core Unit (University Medical Center Göttingen) on a Novaseq system (Illumina). Paired-end sequencing reads were merged using PEAR (version 0.9.6) and sequencing reads were filtered using pyBarcode Filter (pyCRAC, v1.4.5). 3′ adaptor sequences were removed using Flexbar (version 3.5.0) and reads containing identical UMIs were collapsed using pyFastqDuplicateRemover (pyCRAC, v1.4.5). Sequencing reads were aligned to the human genome (vGRCh38.29) using STAR (v2.7.10a). Bam files were indexed with igvtools (v2.12.3) and peaks present in both replicates were called with PEAKachu (v0.2.0) using adaptive mode with default settings and without normalization (-n none). The count matrix was further processed in R (version 4.3.2), where the DESeq2 package (v1.42.1) was used for statistical analysis accounting for batch differences. Peaks were considered significantly enriched in THUMPD2-His$_6$-2xFLAG compared to His$_6$-2xFLAG when they had an adjusted p-value < 0.1, baseMean > 100 and log$_2$ fold change > 2. Genomic coordinates of the significantly enriched peaks were exported as a bed file, which was used to extract reads from the original bam files with samtools (version 1.21). The genomic feature of the reads in the filtered bam files were analyzed with FeatureCounts (v2.0.0) using the following settings: -s 1 -O -t exon -M --largestOverlap --fracOverlap 1 --fraction -g gene_type or gene_name. Alternatively, unmerged sequencing reads were processed as above then aligned to the U6 genomic sequence (Gene ID: 26827) with ten additional 'T's at the 3′ end using Bowtie2 (v2.3.5.1). Bedgraph files were generated using BEDTools (v2.27.1) and aligned reads were normalized using reads per million mapped reads (RPM).

## Quantification of U6 snRNA modifications by LC-HRMS

U6 snRNAs were extracted from HCT116 WT or THUMPD2 KO cells using biotinylated antisense oligonucleotides as in ref. 45 with the difference that the extracted small nuclear RNAs were added to the oligonucleotide-conjugated beads and incubated at 70 °C for 30 min before slowly cooling to 30 °C. 7.2 nmoles of U6 snRNA purified from each cell line using a biotinylated antisense oligonucleotide (5′-biotin-TTTTAGTATATGTGCTGCCGAAGCGAGCAC-3′) were digested into nucleosides using nuclease P1, phosphodiesterase and alkaline phosphatase[45]. For nucleoside analyses, samples from $n = 3$ experiments were diluted with water to enable quantification of canonical and modified nucleosides based on their relative concentrations. The samples were analyzed by LC–high resolution MS (HRMS) and nucleosides of interest identified based on their specific retention time and $m/z$ value: A: 3.5 min, $m/z$ 268.1040; G: 3.7 min, $m/z$ 284.0989; C: 1.4 min, $m/z$ 244.0928; m$^6$A: 4.4 min, $m/z$ 282.1196; Am: 4.2 min, $m/z$ 282.1196; m$^2$G: 4.5 min, $m/z$ 298.1146; Gm: 4.4 min, $m/z$ 298.1146; Cm: 3.4 min, $m/z$ 258.1084. The experiments were performed with a 1260 Infinity II Prime LC system (Agilent Technologies) coupled to a timsTOF high-resolution mass spectrometer (Bruker, Bremen, Germany). The column (CORTECS T3, 120 Å, 2.7 µm, 50 × 2.1 mm; Waters, USA) was chosen for an efficient separation of nucleosides without the use of any buffer such as ammonium acetate in the mobile phase. The elution gradient started from 100% buffer A (water + 0.1% formic acid) to 30% buffer B (acetonitrile + 0.1% formic acid) in 10 min then to 50% buffer B in another 1 min, maintaining at 50% buffer B for 2 min, then going back to 100% buffer A in 1 minute and maintaining this setting for 4 min to equilibrate the column back to initial conditions. The flow rate was fixed at 0.4 mL/min. The injected volume was fixed at 2 µL. For electrospray (ESI) analysis, mass spectra were recorded in positive ion mode with the following parameters: gas temperature 250 °C, drying gas flow rate 10 L min$^{-1}$, nebulizer pressure 3.0 bar, capillary voltage 4200 V, collision cell energy 5.0 eV in MS mode.

For ESI, external calibration was achieved with ESI-L low concentration tuning mix (Agilent, G1969-85000). Standards were purchased from the following vendors: A (Sigma-Aldrich, #A9251), G (Sigma-Aldrich; #G6752), C (Sigma-Aldrich, #C122106), m$^6$A (Berry & Associates, #PR-3732), Am (Berry & Associates, #PR3734-C001), m$^2$G (Sigma-Aldrich; #M4004), Gm (Berry & Associates, #PR3760-C001), Cm (Berry & Associates, #PR7636-C001) and resuspended in milliQ water. Standards were first analyzed by LC–HRMS to determine their elution time, then 10 to 2000 fmol (depending on standards) were injected in triplicate for calibration curves, and peak surface area as a function of standard concentration were plotted for each nucleoside.

## 3′ RACE

For 3′ RACE experiments, HEK293 cells over-expressing THUMPD2-His$_6$-2xFLAG or His$_6$-2xFLAG were UV-crosslinked and used for anti-FLAG immunoprecipitation as described for the CRAC experiments. The eluate from the anti-FLAG affinity purification was supplemented with 6 M guanidine hydrochloride, 300 mM NaCl and 10 mM imidazole pH 8 and purified via the His$_6$-tag under denaturing conditions. Briefly, Ni-NTA beads (Qiagen) were equilibrated in wash buffer 1 (50 mM Tris−HCl pH 7.8, 300 mM NaCl, 0.1% (v/v) NP-40, 6 M guanidine hydrochloride, 10 mM imidazole and 5 mM β-mercaptoethanol) and incubated with the eluate for 2 h at 4 °C. Beads were then washed 3x with wash buffer 1 and 2x with wash buffer 2 (50 mM Tris−HCl pH 7.8, 50 mM NaCl, 0.1% (v/v) NP-40, 10 mM imidazole pH 8 and 5 mM β-mercaptoethanol). RNA was eluted from beads with wash buffer 2 supplemented with 1% sodium dodecyl sulfate (SDS), 5 mM EDTA and 0.325 mg/ml of Proteinase K (Roche) overnight at 55 °C shaking at 650 rpm. Eluted RNA was extracted using phenol:chloroform:isoamylalcohol (25:24:1, v/v) and precipitated using ethanol. For the removal of cyclic 2′,3′-phosphates, the recovered RNA was incubated with 10 U T4 polynucleotide kinase (Thermo Fisher Scientific) in 1x phosphatase removal buffer (100 mM imidazole pH 6, 10 mM MgCl$_2$, 10 mM β-mercaptoethanol and 20 µg/ml BSA) in the presence of 1.6 U/µL Ribolock (Thermo Fisher Scientific) for 3 h at 37 °C. Treated RNA was purified using RNA Clean & Concentrator columns (Zymo Research) and 3′ ends were polyadenylated using a Poly(A)-tailing kit (Thermo Fisher Scientific) for 30 min at 37 °C. Poly-adenylated RNA was re-purified, denatured at 65 °C in the presence of 2 pmol RTQ primer (Supplementary Data 9) and 2 mM dNTPs (Thermo Fisher Scientific) and used for reverse transcription with SuperScript III Reverse Transcriptase (Thermo Fisher Scientific) in the presence of 1x First-strand buffer (Thermo Fisher Scientific), 5 mM DTT, and 1 U/µL Ribolock (Thermo Fisher Scientific) for 50 °C for 1 h. The poly-adenylated U6 snRNA sequence was PCR amplified from the cDNA with primers (Supplementary Data 9) containing 5′ and 3′ linker sequences specific for Illumina adaptors using KOD One™ PCR Master Mix (Sigma-Aldrich). PCR products were gel purified and used for a second PCR reaction with Phusion Hot Start II DNA Polymerase (Thermo Fisher Scientific) with specific primers (Supplementary Data 9) to introduce Illumina adapter sequences with indices for multiplexing. The resulting PCR products were purified using AMPure XP beads (Beckman Coulter), separated by agarose gel electrophoresis and eluted from the gel using the MinElute PCR Purification Kit (QIAGEN). The concentration of the purified PCR products was determined using the Qubit fluorometer (Thermo Fisher Scientific) and deep sequencing was performed as described for the CRAC experiments. FASTQ files were quality checked with FastQC (v0.11.4) and identical reads from the forward sequencing reactions were collapsed using pyFastqDuplicateRemover (pyCRAC version 1.5.2). The UMI and the poly(A) sequence at the 3′ end of the reads were removed using Perl (v5.32.1) and the trimmed sequences were mapped to the U6 snRNA genomic sequence (Gene ID: 26827) extended with 10 extra 'T's at the 3′ end. Alignment was performed using Bowtie2 (v2.5.3), aligned reads were sorted and files were converted to bam and indexed using samtools (sort -O bam

-o and index $input_file.bam) (v1.19). The coverage was generated using bamtools (bamtools coverage -in) (v2.5.2) and obtained read counts were normalized by RPM (reads per million mapped reads).

## UV crosslinking and immunoprecipitation experiments

HEK293 cells over-expressing $His_6$-2xFLAG or THUMPD2-$His_6$-2xFLAG (and its variants) were washed with 1x phosphate-buffered saline (PBS) and crosslinked using UV light at 254 nm (800 mJ/cm²) using a Strata-linker (Stratagene). Cell pellets were resuspended in TMN150 buffer (50 mM Tris-HCl pH 7.8, 150 mM NaCl, 1.5 mM $MgCl_2$, 0.1% NP-40 (v/v) and 5 mM β-mercaptoethanol) supplemented with 1x cOmplete protease inhibitor cocktail (Roche) and lysed by sonicating for 15 cycles at 40% amplitude. Lysates were centrifuged at 20,000 xg for 15 min at 4 °C and the clear lysates were incubated for 2 h at 4 °C with Anti-FLAG M2 magnetic beads (Sigma-Aldrich) pre-equilibrated in TMN150 buffer. The beads were washed 2x with TMN150 buffer and 2x with TMN1000 buffer (50 mM Tris-HCl pH 7.8, 1 M NaCl, 1.5 mM $MgCl_2$, 0.1% NP-40 (v/v) and 5 mM β-mercaptoethanol). Proteins and protein-RNA complexes bound to beads were eluted overnight at 4 °C using 150 μg/ml 3x FLAG peptide (Sigma-Aldrich) in protein elution buffer (50 mM Tris-HCl pH 7.8, 2 mM $MgCl_2$, 0.1% NP-40 (v/v) and 5 mM β-mercaptoethanol) supplemented with 1x cOmplete protease inhibitor cocktail (Roche).

For analysis of proteins by mass spectrometry, eluted proteins/protein complexes were treated with 625 U of benzonase (Sigma-Aldrich) and incubated at 37 °C shaking for 3 h. Samples were precipitated with 15% TCA and pellets were washed with acetone multiple times before being dried in a SpeedVac. For MS analysis, peptides were solubilized in 20 μl of 2% acetonitrile (ACN), 0.05% trifluoroacetic acid (TFA). Alternatively, for analysis of proteins by western blotting, non-benzonase-treated eluates were used directly.

To detect co-purified RNAs, protein-RNA complexes in the anti-FLAG IP eluates were further purified via the $His_6$-tag under denaturing conditions as described in the 3' RACE protocol. After extraction of the RNA with phenol:chloroform:isoamylalcohol (25:24:1, v/v) and precipitation with ethanol, recovered RNAs were analyzed by northern blotting (see section Northern blotting for details).

## IP-MS data analysis

LC–ESI-MS/MS analyses of peptides were performed on an Orbitrap Exploris 480 (Thermo Fisher Scientific) instrument coupled to a nanoflow ultrahigh pressure liquid chromatography system (UltiMate3000 RSLCnano, Thermo Fisher Scientific). The uHPLC column setup was as follows: a trap PepMap Neo C18 5 μm 300 μm x 5 mm cartridge (Thermo Scientific) and a custom in-house packed 34 cm C18 main column (a capillary with 75 μm inner diameter packed with ReproSil-Pur 120 C18-AQ beads, 3 μm pore size, Dr. Maisch GmbH). Samples were separated using a 60-minute method at a flow rate of 300 nl/min using 0.1% (v/v) FA (buffer A) and 80% (v/v) ACN, 0.08% (v/v) FA (buffer B). A two-step linear gradient was applied: first, 10-36% B for 40 min, followed by 3 min of 36–52% B. Eluting peptides were measured in positive mode using a data-independent method with 40 variable m/z windows covering a range of 350–1650. Resolution was set to 120,000 (MS1) and 30,000 FWHM (MS2); AGC targets, 300% (MS1) and 1000% (MS2); normalized collision energy, 30%; maximum injection time, 20 (MS1) and 54 ms (MS2). Four biological replicates were measured with two replicate injections (2 μl per injection).

The raw MS files were processed using the Spectronaut software (v18.3, Biognosys) and the MS/MS spectra were searched against a database containing a UniProt human reference proteome (release 2024-01-24) as well as sequences of the tagged THUMPD2 and of common contaminants found in MS experiments. Protein identifications passing thresholds of 1% FDR at the protein level and a $q$ value < 0.01 were selected for further data processing using R (v4.4.0). After removal of decoy hits and potential contaminant entries, only hits with observations across all four replicates of the THUMPD2-$His_6$-2xFLAG samples were considered for differential abundance analysis. MS2-intensity based abundance of peptides (PG.Quantity) were $\log_2$ transformed and used for the calculation of $\log_2$ fold change (THUMPD2-$His_6$-2xFLAG over $His_6$-2xFLAG). $p$ values were calculated using a Welch two-sided t-test and filtered by $p$ values < 0.05. For data visualization in R (v4.4.0), ggplot2 (v3.5.2) and pheatmap (v1.0.13) packages were used. Only proteins with a predicted nuclear subcellular localization documented in the Human Protein Atlas (release version 24.0) were displayed in the plots.

## Western blotting

For sample preparation, cells were lysed in RIPA buffer (150 mM NaCl, 1% NP-40, 0.5% sodium deoxycholate, 0.1% SDS, 50 mM Tris pH 8, 5 mM $MgCl_2$) supplemented with 1x cOmplete protease inhibitor cocktail (Roche). For immunoprecipitation experiments or preparation of nuclear extracts, cells were lysed as otherwise indicated. Proteins from clear cell lysates were precipitated by addition of trichloroacetic acid (TCA) to 12.5 or 15 % and washed 2x with cold acetone. Protein pellets were resuspended in 2x loading buffer (250 mM Tris-HCl pH 6.8, 40% glycerol, 8% SDS, 5% β-mercaptoethanol and 0.04% bromophenol blue). Equal amounts of protein were separated by SDS-polyacrylamide gel electrophoresis (SDS-PAGE) and transferred to Immobilon-FL PVDF membranes (Merck) or nitrocellulose membranes (Cytiva and Bio-Rad). The membranes were incubated with blocking solution (5% milk in Tris-buffered saline with 0.1% Tween 20 Detergent (TBS-T)). After blocking, membranes were incubated with primary antibodies (Supplementary Data 11) diluted in blocking solution for 2 h at room temperature or overnight at 4 °C. After washing in TBS-T, membranes were incubated with IRDye LI-COR secondary antibodies (Supplementary Data 11) in blocking solution for 1 h at room temperature. After final washing steps, membranes were scanned using an Odyssey CLx scanner (LI-COR Biosciences). Images were analyzed using Image Studio (v5.2.5, LI-COR).

## Fluorescence microscopy and immunofluorescence

HEK293 cells over-expressing GFP-tagged THUMPD2 variants (Supplementary Data 10) were grown on coverslips coated with 2% gelatin (w/v) (Sigma-Aldrich) and fixed with 4% paraformaldehyde (Sigma-Aldrich) in PBS for 20 min at room temperature (RT). Coverslips were washed 3x with 1x PBS and cells were permeabilized with 0.5% Triton-X in PBS for 5 min. After washing, cells were blocked with 0.3% albumin V (Panreac AppliChem) for 20 min and incubated in blocking solution with the primary antibodies listed in Supplementary Data 11 for 1 h at RT. Coverslips were washed 3x with PBS and incubated with Alexa Fluor 594-conjugated secondary antibodies (Supplementary Data 11) for 1 h at RT. After washing, coverslips were mounted onto glass slides using Vectashield (Vector labs) mounting medium supplemented with 1 μg/ml 4′,6-diamidino-2-phenylindole (DAPI) (Sigma-Aldrich). Fluorescence was detected using a Leica Stellaris 5 confocal microscope equipped with a 63x objective using the Leica Application Suite X (v4.6.1.27408). Acquired images were processed using Fiji (version 2.14.0).

For quantification of GFP-THUMPD2 signal in nuclear bodies relative to the nucleoplasm, nuclear bodies were identified with the respective markers (UTP14A or SART3) and a cross-section of a determined length of 3 μm for Cajal bodies or 6 μm for nucleoli was drawn centralized over several nuclear bodies using ImageJ (v2.14.0). The grey value distribution (pixel intensity) in the green channel was obtained using the "Plot Profile" plugin in ImageJ. Values obtained in the middle of the nuclear body cross-section (a distance of 0.5 μm for Cajal bodies or 1 μm for nucleoli) were averaged, along with cross-section values in the surrounding nucleoplasmic area.

## Crosslinking and MS for mapping protein–RNA contact sites

HEK293 cells over-expressing THUMPD2-$His_6$-2xFLAG were washed in 1x PBS and crosslinked using UV or PAR (photoactivatable-ribonucleoside-

enhanced) crosslinking methods in a Stratalinker (Stratagene; $n = 1$ experiment for each type of crosslinking). For UV-crosslinking, cells were exposed to light at 254 nm with 2400 mJ/cm². For PAR-crosslinking, cells were grown in media supplemented with 100 μM 4-thiouridine (Sigma-Aldrich) for 6 h prior to crosslinking at 365 nm with 360 mJ/cm². Crosslinked cells were lysed by sonicating in IP buffer (50 mM Tris−HCl pH 7.4, 150 mM NaCl, 0.5 mM EDTA, 0.1% (v/v) Triton X-100 and 10% (v/v) glycerol) supplemented with 1x cOmplete protease inhibitor cocktail (Roche). Cleared lysates were added to Anti-FLAG M2 magnetic beads (Sigma-Aldrich) pre-equilibrated in IP buffer. After incubating for 3 h at 4 °C, the beads were washed five times with IP buffer. Proteins and protein−RNA complexes bound to beads were eluted using 250 μg/ml of 3x FLAG peptide (Sigma-Aldrich) in 1x TBS for 1 h at 4 °C. Eluates were precipitated using 3 volumes of ethanol and further sample processing was performed as described before[95]. Protein−RNA complexes were dissolved in 4 M urea and 50 mM Tris-HCl pH 7.5 by sonication, then diluted four-fold with 50 mM Tris-HCl pH 7.5. 500 U Pierce Universal Nuclease (Thermo Fisher Scientific) and 200 U nuclease p1 (NEB) were added and incubated at 37 °C for 2 h. 10 μg RNase A (EN0531, Thermo Fisher Scientific) and 1 U RNase T1 (Thermo Fisher Scientific) were added and samples were incubated for a further 2 h at 37 °C. Proteins were digested using 4 μg trypsin (Promega) at 37 °C overnight. Samples were purified twice using C18 micro spin columns (Harvard Apparatus) according to the manufacturer's instructions and eluates were dried in a SpeedVac concentrator. Crosslinked peptides were enriched using TiO₂ columns (in-house; Titansphere 5 μm; GL Sciences), as described before[96]. Samples dissolved in buffer A (5% (v/v) glycerol, 80% (v/v) acetonitrile (ACN), 5% (v/v) trifluoroacetic acid (TFA)) were loaded onto equilibrated TiO₂ columns. Columns were washed three times with buffer A, three times with buffer B (60% (v/v) ACN, 5% (v/v) TFA) and once with 60% (v/v) ACN, 0.1% (v/v) TFA. Samples were eluted using ammonia solution and eluates were dried in a SpeedVac concentrator before LC-ESI-MS/MS analysis.

Enriched (oligo)nucleotide crosslinked peptides were dissolved in 2% (v/v) ACN, 0.05% (v/v) TFA. LC−MS / MS analyses were performed on an Orbitrap Exploris 480 (Thermo Fisher Scientific) instrument coupled to a nanoflow liquid chromatography system (UltiMate™3000 RSLCnano, Thermo Fisher Scientific). Sample separation was performed over 58 min at a flow rate of 300 nl/min using 0.1% (v/v) FA (buffer A) and 80% (v/v) ACN, 0.08% (v/v) FA (buffer B) and a linear gradient from 10% to 45% buffer B in 44 min. Eluting (oligo)nucleotide crosslinked peptides were analyzed in positive mode using a data-dependent top 20 acquisition method. Resolution was set to 120 000 (MS1) and 30 000 FWHM (MS2); scan range ($m/z$), 350–1600; AGC targets, 1e6 (MS1) and 1e5 (MS2); normalized collision energy, 28%; dynamic exclusion, 9 s; maximum injection time, 60 (MS1) and 120 ms (MS2). Measurements were performed in two replicate injections. MS data were analyzed and manually validated using the OpenMS pipeline NuXL (v1)[97,98] and OpenMS TOPPViewer (v2.6.0). Crosslink sites are reported at 1% spectrum level FDR.

## Structure modeling
Models of the human THUMPD2-TRMT112 complex were obtained using AlphaFold (v3)[57]. Among the five models proposed by the AlphaFold server (https://alphafoldserver.com/) using default settings (seed:auto), the best had a ipTM score of 0.89. All five models were very similar with root mean square deviation (rmsd) values lower than 0.5 Å. In all these models, the interface between TRMT112 and THUMPD2 was similar to those experimentally observed in the complexes between eukaryotic TRMT112 proteins and known MTase partners[62,99]. For manual modeling of a THUMPD2−TRMT112−U6 complex, the high-resolution crystal structure of the *S. cerevisiae* core U6 snRNA as observed in complex with four RRM domains of Prp24 was used[100] (PDB: 4N0T). As in this structure, the ISL is not visible due to intrinsic flexibility, the cryo-EM structure of the human activated

spliceosome precursor (pre-B$^{act}$; PDB: 7ABI)[101] was used to generate a hybrid model of the U6 snRNA containing the ISL. This model lacks the mature 5′- (nucleotides upstream of A$_{24}$) and 3′- (nucleotides downstream C$_{94}$) ends. To generate the model of U6 bound to the THUMPD2-TRMT112 complex, the crystal structure of ThiI, another THUMP containing enzyme, bound to the amino acyl acceptor arm of a tRNA (PDB: 4KR7)[54] was superposed onto the AlphaFold model of the THUMPD2−TRMT112 complex using THUMP domains as reference. This allowed modeling of a double-stranded RNA bound to the THUMPD2 THUMP domain. This double-stranded RNA served as a reference to superpose the ISL of the hybrid model of U6 snRNA described above.

## Chemical synthesis of RNA oligonucleotides
Unmodified and 2′-O-methylated RNA oligonucleotides were purchased from IDT or prepared by solid-phase synthesis in the Höbartner lab, and m²G-containing RNA oligonucleotides were generated by solid-phase synthesis (Höbartner lab; Supplementary Data 12). Solid phase synthesis was performed on a K&A DNA/RNA synthesizer H6/H-8. The synthesis was done with a 1000 Å controlled-pore glas (CPG) support (ChemGenes, 20–25 μmol/g, 1 μmol synthesis scale). 2′-O-TOM-protected $N^6$-acetyl-rA, $N^4$-acetyl-rC, $N^2$-acetyl-rG, and rU phosphoramidites were procured from ChemGenes and prepared as 70 mM solutions in dry acetonitrile. 5′-O-DMT-2′-O-TOM-O$^6$-NPE-m²G 3′-CEP phosphoramidite was synthesized as reported by Wachowius[102] and used as 100 mM solution in dry acetonitrile. Phosphoramidite solutions were prepared in dry acetonitrile (100 mM for unmodified, 70 mM for modified building blocks) and coupled for 4 min (canonical nucleosides) or 10 min (modified phosphoramidites) upon activation with 0.25 M ethylthiotetrazole (ETT). Detritylation was performed with 3% dichloroacetic acid (DCA) in dichloromethane (DCM). Capping solutions contained THF/lutidine/Ac₂O 8/1/1 (v/v/v) and THF/$N$-methylimidazole (NMI) 84/16 (v/v). Oxidation was done with 50 mM I₂ in pyridine/H₂O 9:1 (v/v). Afterwards, cleavage from the solid support and alkaline deprotection was performed with MeNH₂/NH₃ (1:1) in water (AMA) for 6 h at 37 °C. TOM-protecting groups were removed with 1 M TBAF in dry THF for 12 h. The crude RNA oligonucleotides were desalted by size exclusion chromatography (GE Healthcare HiTRAP desalting columns (3 × 5 ml), flow rate 2 ml/min at ambient temperature) and purified as described in the section for preparation of NMR samples.

## DNAzyme-catalyzed RNA cleavage
Total RNA was isolated from HCT116 WT and THUMPD2 KO cell lines untreated or after treatment with siRNAs (Supplementary Data 13) using TRI Reagent (Sigma-Aldrich) according to the manufacturer's instructions. RNA concentrations and purity were determined on a Nanodrop One (Thermo Fisher Scientific). For each reaction, 2.5 μg of total RNA were mixed with 100 pmol of the appropriate DNAzyme and 100 pmol of disruptor oligonucleotides (Supplementary Data 9) in the appropriate kinetic buffer (50 mM Tris pH 7.5, 150 mM NaCl for U6-m²G$_{72}$/U6atac-G$_{44}$ detection, 50 mM HEPES pH 7.4, 400 mM KCl and 100 mM NaCl for U6-Am$_{70}$ detection or 50 mM HEPES pH 7.4 and 150 mM NaCl for U6-Cm$_{62}$/Cm$_{63}$ detection). The reactions were denatured at 95 °C for 4 min and cooled down to 25 °C at a rate of 0.1 °C/s. After cooling, reactions were supplemented with Ribolock (Thermo Fisher Scientific) to a final concentration of 1 U/μl and MgCl₂ and/or MnCl₂ (5 mM MgCl₂ for U6-m²G$_{72}$/U6atac-G$_{44}$ detection, 10 mM MgCl₂ and 10 mM MnCl₂ for U6-Am$_{70}$ detection, and 1 mM MgCl₂ and 1 mM MnCl₂ for U6-Cm$_{62}$/Cm$_{63}$ detection). The reactions were incubated for 5 h or 6 h at 37 °C (m²G detection) or 25 °C (2′-O-methylation detection) before quenching by addition of 2x RNA loading dye (95% formamide, 0.5 mM EDTA, 0.025% bromophenol blue, 0.025% xylene cyanol, 0.025% SDS). To analyze the formation of cleavage products, samples were heated at 85 °C for 5 min and 500 ng of treated total RNA

were resolved in 8 or 10% denaturing (7 M urea) polyacrylamide gels and northern blotted.

## Northern blotting

After separation by denaturing PAGE, RNAs were transferred to a Hybond-N membrane (Cytiva) and UV-crosslinked 2x at 120 mJ/cm$^2$ using a Stratalinker (Stratagene). Membranes were incubated in pre-hybridization solution (250 mM sodium phosphate pH 7.4, 7% SDS and 1 mM EDTA pH 8.0) for 30 min at 37 °C and hybridized with [$^{32}$P]-labelled DNA oligonucleotides (Supplementary Data 9) in the same buffer at 37 °C overnight. Membranes were then washed sequentially with 6x saline sodium citrate (SSC) buffer and 2x SSC buffer supplemented with 0.1% SDS for 30 min each at 37 °C. Membranes were exposed to phosphorimager screens and signals were detected using a Typhoon FLA 9500 (GE Healthcare Life Sciences) with the Typhoon FLA9500 Control Software (v1.0) or an Amersham Typhoon 5 (Cytiva) using ImageQuant (v8.1). Bands were quantified where appropriate using ImageJ (v2.14.0) or Image Studio (v5.2.5, LI-COR).

## Recombinant protein expression in *E. coli* and protein purification

For the heterologous expression of the wild type *Bos taurus* (*Bt*) THUMPD2-TRMT112/*Bt*THUMPD2-TRMT112 complexes in *E. coli*, a pET28 vector (Supplementary Data 10) was used for co-expression of codon optimized His$_6$-ZZ tagged *Bt*THUMPD2 and untagged *Bt*TRMT112[45]. *E. coli* BL21(DE3) Codon+ were transformed with the aforementioned plasmid and grown in TB media (24 g/L yeast extract, 12 g/L tryptone, 4 ml/L glycerol, 0.17 M KH$_2$PO$_4$, 0.72 M K$_2$HPO$_4$) supplemented with appropriate antibiotics at 37 °C to OD$_{600}$ = 0.6. Co-expression of the THUMPD2–TRMT112 complex was achieved in an overnight culture at 18 °C after addition of 0.5 mM isopropyl β-D-1-thiogalacto-pyranoside (IPTG). Harvested cells were resuspended in cold lysis buffer (20 mM Tris–HCl pH 8, 200 mM NaCl, 5 mM β-mercaptoethanol, 0.2 mM PMSF) and lysed at 4 °C using an Emulsiflex-C3 (Avestin). The lysates were cleared by centrifuging at 20 000 xg for 30 min and the soluble proteins were incubated for 1 h at 4 °C with cOmplete His-Tag Purification Resin (Roche) pre-equilibrated in lysis buffer. The resin was washed once with lysis buffer supplemented with 20 mM imidazole pH 8 and wash buffer (20 mM Tris–HCl pH 8, 1000 mM NaCl, 5 mM β-mercaptoethanol) to remove unspecific bound proteins. The retained proteins were eluted with elution buffer (20 mM Tris–HCl pH 8, 200 mM NaCl, 5 mM β-mercaptoethanol, 400 mM imidazole pH 8) and dialyzed overnight against a buffer containing 20 mM Tris–HCl pH 8, 50 mM NaCl and 5 mM β-mercaptoethanol (buffer A). The dialyzed proteins were injected onto a Mono Q 5/50 GL column (Cytiva) for ion-exchange chromatography purification. The proteins were eluted using a gradient of NaCl from 50 mM (buffer A) to 1 M NaCl (buffer B). Eluted proteins were dialyzed overnight in dialysis buffer (20 mM Tris–HCl pH 8, 200 mM NaCl, 5 mM and 50% glycerol). For fluorescence anisotropy experiments, proteins were dialyzed in dialysis buffer containing 20% glycerol, snap frozen and stored at -80 °C. Purified proteins were examined using Coomassie G-250-stained SDS-polyacrylamide gels and protein concentrations were determined using the Pierce BCA Protein Assay Kit and Pierce Coomassie Plus (Bradford) reagent (Thermo Fisher Scientific).

## In vitro transcription

For in vitro transcription of the U6 snRNA, reactions were prepared using 2 μg of BsaI-linearized plasmids (Supplementary Data 10), 1 mM each NTP (ATP, UTP, CTP, GTP; Thermo Fisher Scientific), 1x transcription buffer (Thermo Fisher Scientific), 1 U/μL Ribolock (Thermo Fisher Scientific) and 2 μg of homemade T7 RNA polymerase in a 50 μL volume reaction. Reactions were incubated for 3 h at 37 °C and stopped by addition of 2 U of Turbo DNAse I (Thermo Fisher Scientific) and incubation for further 30 min at 37 °C. Transcripts were column-

purified using the RNA Clean & Concentrator kit (Zymo Research), according to the manufacturer's instructions and refolded by denaturing at 80 °C for 2 min followed by slow cooling to room temperature. The RNA concentration and purity were determined using a Nanodrop One$^c$ (Thermo Fisher Scientific) and integrity of the transcripts was verified by analysis on denaturing 7 M urea polyacrylamide gels stained with SYBR Gold (Thermo Fisher Scientific).

## In vitro methylation assays

For in vitro methylation assays, 20 μL reactions were prepared containing 2 μCi of *S*-[methyl-$^3$H]-adenosyl-L-methionine ([$^3$H]-SAM; Hartmann Analytic (Fig. 4c–e, m and n) or Revvity (Fig. 4f, p and s), 5 μM of recombinant His$_6$-ZZ-*Bt*THUMPD2-*Bt*TRMT112 complex in 1x methylation buffer (50 mM Tris-HCl pH 7.4, 50 mM NaCl, 5 mM MgCl$_2$, 1 mM DTT) supplemented with 1 U/μL RiboLock RNase Inhibitor (Thermo Fisher Scientific). Reactions were incubated for 10 min at 25 °C, then mixed with 20 pmol of synthetic oligonucleotides (Supplementary Data 12) or in vitro transcribed snRNA. Methylation reactions were incubated at 25 °C for 2 h and stopped by adding Proteinase K (Roche) to a final concentration of 1 mg/ml. After 30 min incubation at 25 °C, RNAs were ethanol precipitated overnight at -20 °C. Samples were centrifuged at 20 000 xg for 30 min, and RNA pellets were washed with 70% ethanol and resuspended in nuclease-free water (QIAGEN). Incorporation of [$^3$H] in the RNA was measured using scintillation counting in a Hidex 300SL (HIDEX) using MikroWin 300 SL (v5.63).

## RNAi-mediated depletion of LARP7

HCT116 WT and THUMPD2 KO cells were seeded at a density of 2-3×10$^5$ cells/well (WT) or 2.2-3.5×10$^5$ cells/well (THUMPD2 KO) in 6-well dishes. Wells with lower cell densities were transfected with non-target siRNAs (40 nM; Supplementary Data 13) and wells with higher cell densities were transfected with siRNAs against LARP7 (40 nM; Supplementary Data 13) using Lipofectamine RNAiMAX reagent (Thermo Fisher Scientific) according to the manufacturer's instructions. Cells were harvested 72 h after transfection and extracted proteins were used for western blotting, whereas extracted RNA was used for primer extension assays, DNAzyme-catalyzed cleavage of RNA and RNA-seq experiments. For depletion of LARP7 for the preparation of nuclear extracts, HCT116 WT and THUMPD2 KO cells were seeded at a density of 2.6 ×106 cells/15-cm dish (WT) or 3 ×10$^6$ cells/15-cm dish (THUMPD2 KO).

## U6 sequence alignment

The sequences of U6 snRNAs from different species were retrieved from Genbank (http://www.ncbi.nlm.nih.gov/genbank/; Release 260) and were aligned with MAFFT (v7) using standard parameters.

## Fluorescence anisotropy

For fluorescence anisotropy measurements, His$_6$-ZZ tagged *Bt*THUMPD2 and untagged *Bt*TRMT112 complex were dialyzed against anisotropy buffer (20 mM Tris-HCl pH 8, 50 mM NaCl and 5 mM β-mercaptoethanol) overnight at 4 °C. Reactions containing 0-10 μM protein complex and 20 nM 6-FAM-labelled oligos (IDT; Supplementary Data 12) in anisotropy buffer were measured at 22 °C in a FluoroMax-4 spectrofluorometer (Horiba Scientific) in a quartz glass cuvette (Hellma Analytics) using FluorEssence Software (v3.9). The fluorophore was excited at 495 nm and emission was measured at 520 nm. The binding affinity of the protein complex towards the substrate RNAs (total anisotropy or A$_T$) was calculated using Eq. (1):

$$A_T = A_R$$

$$+ \frac{(A_{PR} - A_R)}{[\text{RNA}]_T} \cdot \left( \frac{[\text{protein}]_T + [\text{RNA}]_T + K_d}{2} - \sqrt{\left( \frac{[\text{protein}]_T + [\text{RNA}]_T + K_d}{2} \right)^2 - [\text{protein}]_T [\text{RNA}]_T} \right)$$

$$(1)$$

where $A_R$ is the anisotropy of unbound RNA, $A_{PR}$ is the anisotropy of the heteroprotein bound to RNA, and $[protein]_T$ and $[RNA]_T$ are the total concentrations of protein and RNA, respectively.

## mRNA-seq and alternative splicing analysis

Total RNA was extracted from HCT116 WT or THUMPD2 KO cells after treatment with NT siRNA or siRNAs against LARP7 (Supplementary Data 13) using TRI reagent (Sigma-Aldrich) according to the manufacturer's instructions. To remove possible genomic DNA contamination, RNA samples were treated with TURBO DNase (Invitrogen) for 15 min at 37 °C and cleaned up using the RNA Clean & Concentrator kit (Zymo) according to the manufacturer's protocol. Polyadenylated RNAs were enriched and library preparation was performed with the TruSeq Stranded Total RNA kit (Illumina) according to standard protocols. Paired-end 50 bp sequencing was conducted using a Novaseq. Library preparation, quality analyses and sequencing were done by the NGS Integrative Genomics (NIG) Core Unit of the University Medical Center Göttingen (UMG).

For mapping, the aligner STAR (v2.7.11b) with the following settings was used: –outFilterMultimapNmax 10 –outSAMattributes All –outSAMtype BAM SortedByCoordinate –outReadsUnmapped Fastx –chimSegmentMin 20 –chimOutType WithinBAM Junctions –quant Mode TranscriptomeSAM GeneCounts –outWigType bedGraph –out WigNorm RPM outWigStrand Unstranded –outFilterMismatchNmax 2[103]. UCSC Genome Browser tools bedSort (v469) and bed-GraphToBigWig (v469) were used to convert from bedgraph format to bigwig format for visualization of the data in the IGV genome browser[104,105]. Genome assembly version GRCh38.p14 and the GEN-CODE gene set from release 47 were used in gtf- or converted to bed-format. For visualization of alternative splicing events as sashimi plots (Fig. 5c-d) bam files of biological replicates were merged and indexed using SAMtools (v1.9). Gene expression was quantified using the DESeqDataSetFromMatrix, DESeq, results and plotPCA functions from the DESeq2 package (v1.48.2) in Bioconductor[106]. To obtain read counts per gene, bam files were split into two files containing either forward or reverse reads using SAMtools (v1.9) and intersected with the gene annotation using BEDTools intersect BEDTools (v2.27.1) requiring at least 3 nucleotides in overlap, a genomic distance between read start and end smaller than then feature length and mapping to the strand of the annotated feature (or the opposite strand for the reverse read)[107,108]. Reads overlapping with different features were counted using awk and summed between overlaps originating from forward and reverse reads. Only genes with 50 or more reads were retained for downstream analysis and plotting. Principal component analysis was based on the top 5000 genes. Differential expression results are listed in Supplementary Data 6.

For the analysis of alternative splicing (AS), rMATS-turbo (v4.3.0) for unpaired replicates was used[109]: rmats.py --b1 R1.bam --b2 R2.bam --nthread 4 --gtf 4 annotations/hg38/gencode.v47.chr_patch_hapl_scaff.annotation.gtf --od out --tmp out_prep -t paired --readLength 50 --variable-read-length --libType fr-firststrand --task both --novelSS --mel 6000 --allow-clipping. The maximum exon length (mel) was changed to 6000 to initially include RI (retained intron) events for longer introns. To restrict this increase in mel to the middle exon, which actually reflects the adjacent exons and the intron in between, for RI, all other exons linked to a particular AS event were required to be at most 500 nt long (corresponding to the default mel 500). This ensured that typical exon architecture is met for different AS events and intron retention could be better represented capturing also long introns[110]. Nevertheless, this only reported back on inclusion levels of a minority of introns, as only few introns are annotated as retained in the gtf file, or produced often wrongly assigned introns, as seen by manual inspection. Hence, intron retention (IR) was quantified analogously to the approach taken for calculating splicing efficiencies in FRASER[111] or

the SPI[112] with custom scripts and the intron annotation derived from the Ensembl annotation version 115 in biomaRt 2.64.0 from Bioconductor 3.21.

Specifically, bam files were split into forward and reverse reads again using samtools view -h -b -f 0×40 \$BAM > r1.bam/0×80 \$BAM > r2.bam. Afterwards, reads were overlapped with the annotated introns maintaining strand information using BEDTools intersect. A minimum overlap of three bases was required to consider a read overlapping. The block count shows the number of continuous aligned segments in a read. Reads with a block count of 1, indicating alignment without gaps, were considered unspliced and were further categorized into unspliced or intronic. Reads were classified as intronic if start and end were within start and end of the respective intron. If the block count was > 1, indicating a gap in the aligned read and thus an exon-exon junction, it was classified as spliced. Split reads were further divided into spliced (exactly matching annotated intron) or alternatively spliced (no perfect match). For each intron, reads for each of the categories were counted. For each intron, IR fraction was calculated by dividing the number of unspliced reads by the sum of the number of unspliced and 2x the number of spliced reads. Significance of IR differences was assessed using the two-tailed Student's t-Test. Data mapping and quantification of alternative splicing and expression were done using the curta system provided by the HPC Service of FUB-IT, Freie Universität Berlin (10.17169/refubium-26754).

For all types of AS, a median of 50 reads per splicing event across replicates of at least one condition was required to consider events for further analysis and only junctions previously annotated in the Ensembl database were considered. These filtering steps were implemented in custom R scripts after running rmats.py or the IR quantification, respectively. Furthermore, only events that passed the read cutoff in all conditions were included in the AS comparison: A3SS 2069 events, A5SS 1318 events, MXE 1543 events, SE 12265 events, IR 37885 events. AS events quantified by rMATS were considered significant with an FDR < 0.01, an absolute inclusion difference of at least 0.1 (Supplementary Data 7-8). To call significant IR events an absolute inclusion difference of at least 0.05 and p-value < 0.01 were required. maser (v1.26.0), pheatmap (v1.0.13), ggplot2 (v4.0.1), UpSetR (v1.4.0), ComplexUpset (v1.3.6) and ggsashimi (v1.1.5) packages were used for analysis and data visualization in R (v4.5.2). Other used packages were biomaRt (v2.64.0), GenomicFeatures and GenomicRanges (v1.60.0), AnnotationDbi (v1.70.0), Biobase (v2.68.0), GenomeInfoDb (v1.44.3), IRanges (v2.42.0), tibble (v3.3.0), gridExtra (v2.3), ggpubr (v0.6.2), RColorBrewer (v1.1-3), dplyr (v1.1.4), tidyr (v1.3.1).

Analysis of 5′ and 3′ splice site strength in association to IR was done and visualized as reported previously[45].

To address the contribution of the combined THUMPD2 KO & LARP7 KD towards measured inclusion differences (ID) relative to wild type treated with a non-targeting siRNA, a weighted mean of the inclusion level differences resulting from lack of either THUMPD2 or LARP7 individually was calculated using the following formula: $ID_{THUMPD2\ KO} * (1 - scaling\ factor) + ID_{LARP7\ KD} * scaling\ factor = ID_{calculated\ THUMPD2\ KO\ \&\ LARP7\ KD}$.

To visualize data using boxplots, the default settings of the geom_boxplot function of the ggplot2 package (v4.0.1) were used. In this case, the centre line reflects the median, bounds of the boxes (hinges) reflect the 25th percentile (Q1) and 75th percentile (Q3), respectively. The length of the box is the Interquartile Range (IQR = Q3 - Q1). The whiskers extend from the hinges to the smallest or largest volume that is no further 1.5x IQR from the hinge. Additional points in the plot indicate any data points lying beyond the ends of the whiskers (outliers), the most extreme points replect the minima and maxima, respectively.

## Thermal UV-Vis melting curves

Melting curve studies were performed on a Cary 3500 from Agilent in an Agilent Cary UV-Vis Multicell Peltier. All samples used for thermal melting curve studies contained 10 mM sodium phosphate buffer (pH 7.4) and 100 mM NaCl. For melting curves of the monomolecular U6 snRNA construct, two RNA concentrations were measured (1 μM and 10 μM). For the bimolecular U4/U6 di-snRNP construct, five RNA concentrations were measured (1 μM, 2 μM, 5 μM, 10 μM and 20 μM). Two heating and two cooling ramps between 10 °C and 90 °C with a heating rate of 0.5 °C per minute were recorded.

## 1D $^1$H-NMR spectroscopy

After cleavage from solid phase support and deprotection, RNAs were purified on a 15% denaturing polyacrylamide gel. Vivaspin 2 centrifugal concentrators (Sartorius) with MWCO 3000 were used for desalting. RNA samples were precipitated with 5 volumes of 2% (w/v) $LiClO_4$ in acetone for 6 h at -20 °C. The NMR sample contained RNA (110-120 μM) in 10 mM potassium phosphate buffer (pH 7.4), in a final volume of 180 μl 90% $H_2O$/10% $D_2O$. For internal referencing, 3-(trimethylsilyl)-1-propanesulfonic acid (DSS) was added. Each NMR sample was annealed at 95 °C for 3 min and subsequently stored on ice for at least 1 h. Correct folding of the samples was checked with an analytical 15% PAA PAGE under native conditions. After 1 h at 100 V, the gel was stained with 1x SYBR gold in TBE buffer and imaged using a ChemiDOCMP with UV Trans Illumination and a standard filter.

All spectroscopy experiments were performed on a Bruker Avance III HD 600 MHz NMR spectrometer equipped with a DCH $^{13}$C/$^1$H cryoprobe with z-gradient and ATM. The titration experiments were performed on samples in a 3 mm Tube. Spectra were measured between 288 K and 318 K. Water suppression of the spectra recorded for the imino region was achieved with a jump-and-return scheme, with setting the maximum of excitation to 12.5 ppm (mid of imino region[113]). Otherwise, excitation sculpting was used for water suppression[114]. For data acquiring, processing and analysis TopSpin (v3.5 pl7) or TopSpin (v4.3.0, Bruker BioSpin, Germany) was used.

## snRNP gradients

Nuclear extracts were prepared from HCT116 WT, THUMPD2 KO cells or WT/THUMPD2 KO cells after LARP7 KD (section RNAi-mediated depletion of LARP7) for the enrichment of snRNPs as described previously[115]. Briefly, cells were grown to 80% confluency in 7-10×15 cm dishes, harvested by trypsinization and washed twice with cold PBS. Cells were gently resuspended in five packed cell pellet volumes of hypotonic buffer A (10 mM HEPES pH 7.9, 1.5 mM MgCl$_2$, 10 mM KCl, 0.5 mM DTT, 0.5 mM PMSF supplemented with 1x cOmplete protease inhibitor cocktail (Roche)), allowed to swell on ice for 10 min and centrifuged at 700 xg for 5 min. Cells were resuspended in 6 ml of buffer A and lysed using a glass Dounce homogenizer (Sigma-Aldrich) until it could be detected by Trypan blue staining that >80% of the cells were lysed. The homogenate was centrifuged for 15 min at 700 xg and the nuclear pellet was resuspended in 1 packed cell volume of low-salt buffer (20 mM HEPES pH 7.9, 20 mM KCl, 1.5 mM MgCl$_2$, 0.2 mM EDTA, 25% glycerol (v/v), 0.5 mM DTT and 0.5 mM PMSF) and gently stirred for 30 min at 4 °C after addition of the equivalent volume of high salt buffer (20 mM HEPES pH 7.9, 1.2 M KCl, 1.5 mM MgCl$_2$, 0.2 mM EDTA, 25% glycerol (v/v), 0.5 mM DTT and 0.5 mM PMSF). The homogenate was cleared by centrifugation for 30 min at 9,000 xg and dialyzed twice against 50 volumes of buffer D (20 mM HEPES pH 7.9, 100 mM KCl, 0.2 mM EDTA, 20% (v/v) glycerol, 0.5 mM DTT and 0.5 mM PMSF) for 2 h at 4 °C. After dialysis, the extract was centrifuged at 9,000 xg for 30 min and the clear supernatant was aliquoted, snap-frozen in liquid-nitrogen and stored at −80 °C until use. For the gradients, 200 μL of each nuclear extract was diluted 1:1 in gradient dilution buffer (20 mM HEPES pH 7.9, 100 mM KCl and 1 mM MgCl$_2$) and loaded

on top of a glycerol gradient (20 mM HEPES pH 7.9, 100 mM KCl, 1 mM MgCl$_2$ and 10–30% (v/v) glycerol). snRNP complexes were separated by centrifugation at 34,000 rpm for 18 h at 4 °C in an SW40Ti rotor and fractions were collected using a Biocomp fractionator. RNA extracted from the fractions using TRI reagent LS (Sigma-Aldrich) was analyzed by northern blotting using oligonucleotides in Supplementary Data 9. Proteins precipitated from the fractions by treatment with 15% (trichloroacetic acid) TCA were analyzed by western blotting using antibodies listed in Supplementary Data 11.

## Statistical analyses

For statistical analyses of confocal microscopy data, unpaired t-tests were performed using R (v4.3.1) and dots overlayed on the box plots represent individual data points. Statistical analyses of data from scintillation counting and blot quantifications were performed using the GraphPad Prism software (v9), with error bars representing mean ± standard deviation and dots indicating the individual data points of $n = 3$ independent experiments. Statistical analyses were done using one-way ANOVA for groups of three or more and significance was calculated using Tukey's multiple comparisons test, while for two-sample comparison two-tailed unpaired t-tests were performed. For statistical analyses of mass spectrometric data, a Welch two-sided t-test was used to generate applicable $p$ values using R (v4.3.1). The statistical tests used for alternative splicing analyses are indicated in the relevant figure legends.

## Reporting summary

Further information on research design is available in the Nature Portfolio Reporting Summary linked to this article.

## Data availability

The deep sequencing data generated during this study have been deposited in Gene Expression Omnibus (GEO) database (http://www.ncbi.nlm.nih.gov/geo/) under the following accession codes: GSE270752: CRAC datasets for THUMPD2-His$_6$-2xFLAG and the His$_6$-2xFLAG tag. GSE270753: 3' RACE datasets for THUMPD2-His$_6$-2xFLAG and the His$_6$-2xFLAG tag. GSE303357: RNA-seq datasets for WT and THUMPD2 KO treated with siNT or siLARP7_1. The proteomics data generated during this study have been deposited at the ProteomeXchange Consortium via the PRIDE partner repository (https://www.ebi.ac.uk/pride/) under the following accession code and submission reference: PXD053510: IP-MS datasets for THUMPD2-His$_6$-2xFLAG and the His$_6$-2xFLAG tag. PXD053619: Protein-RNA cross-linking MS dataset for THUMPD2-His$_6$-2xFLAG. Source data are provided with this paper.

## Code availability

The code developed for optimized intron retention analysis is available via GitHub (https://doi.org/10.5281/zenodo.18197506).

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

## Acknowledgements

We thank Nathalie Ulryck for technical assistance, Manuela Michel for resynthesis of $m^2G$ phosphoramidite, and Anam Liaquat for development and initial characterization of deoxyribozymes that cleave pyrimidine-pyrimidine junctions (here applied to C63 | U64). We thank Matthias Dobbelstein for providing the PRPF8 antibody, Nicolás Lemus-Diaz for helpful discussions, and Monika Raabe and Ralf Pflanz for their assistance in MS analysis. We acknowledge the HPC Service of FUB-IT, Freie Universität Berlin for providing the service and computing time. FUNDING This work was supported by the Deutsche Forschungsgemeinschaft (DFG) via SFB1565 (project number 469281184; P04 to H.U., P12 to K.E.B., and P18 to C.H. and M.T.B., associated project L.H.) and BO5699/1-1 to K.E.B., the Centre National pour la Recherche Scientifique (CNRS) and Ecole Polytechnique to M.Graille and the Agence Nationale pour la Recherche (ANR; ANR-22-CE12-0010-01) to M.Graille. L.L.E.T. is the recipient of a PhD track fellowship from the Institut Polytechnique de Paris.

## Author contributions

K.E.B. conceived and coordinated the study, and N.K. and K.E.B. designed experiments. N.K. performed most experiments. J.P. contributed to immunoprecipitation experiments, and P.H. contributed to northern blotting. M.Greve performed thermal melting curves, $^1$H-NMR spectroscopy and developed DNAzymes. P.H. conducted CRAC experiments, and C.C.T. performed bioinformatic analysis of CRAC datasets. L.H. performed bioinformatic analyses of the mRNA-seq datasets. L.M.W. and O.D. performed mass spectrometry analysis, and L.L.E.T. and D.T. performed LC-HRMS analysis. M.Graille modeled the THUMPD2-TRMT112-U6 complex. H.S., M.T.B., H.U., M.Graille, C.H. and K.E.B. supervised the work. N.K. and K.E.B. wrote the manuscript with contributions from L.H., M.G. and C.H. All authors analyzed data and commented on the manuscript.

## Funding

## Competing interests

The authors declare no competing interests.
