## [Transparent Peer Review file · Nature Communications]

2'-O-methylation-dependent installation of N2-methylguanosine in the U6 internal stem loop facilitates efficient spliceosome assembly

Corresponding Author: Dr Katherine Bohnsack

Version 0:

Reviewer comments:

Reviewer #1

(Remarks to the Author)
Kleiber et al.

2'-O-methylation-dependent installation of N2-methylguanine in the U6 internal loop facilitates efficient spliceosome assembly

In this manuscript, the authors have biochemically and functionally characterized the enzyme complex that m2G-modifies the U6 snRNA at position 72. Although this link has been made before, the authors show that this is the only modification that THUMP2 generates. THUMP2 is enriched in Cajal bodies, it associates with the adaptor protein TRMT112 and binds to U6 at later stages of U6 RNP assembly. The authors further use crosslinking experiments to map the interaction of THUMP2 to U6 and TRMT112. U6 is contacted at multiple regions in the N and the C terminal part of the protein, particularly the THUMP and the MTase domain. TRMT112 was mapped to the C terminus of the protein. They further generated a DNAzyme that allows for the site-specific identification of m2G72. THUMP2 knock out cells clearly show reduced modification at this position and complementation assays revealed that SAM binding and TRMT112 binding is required for methylation in cells. However, in vitro, the reaction was not efficient and therefore the authors investigated the contribution of associated 2'-O-methylation events. Indeed, pre-2'-O-methylated substrates were efficiently modified by the reconstituted enzyme complex in vitro. Using LARP7 knock down, which globally reduced U6 2'-O-methylation, the authors further show that prior U6 2'-O-methylation is also required in cells. Finally, the authors find that alternative splicing events are affected by THUMP2 knock out, LARP7 knock down and a stronger effect when both are inactivated simultaneously. This appears to be due to less efficient di- and tri-snRNP formation.

This is a very solid and clear manuscript. All experiments are competently carried out, presented and interpreted. Although the link between THUMP2 and U6 m2G72 has already been made, the manuscript goes far beyond and presents a thorough characterization of the entire pathway including experiments in cells, generation of new tools such as DNAzymes as well as biochemical in vitro reconstitution of the enzymatic activities. I have only a few comments that could be considered.

1. Fig. 1A: about 60% of the reads are crosslinked to lncRNAs or mRNAs (which is also seen in the negative control). It is clear that no modification sites besides the one in the ISL of U6 were found. However, THUMP2 may bind to other transcripts without methylating them. Have some of the mRNAs or lncRNAs been validated to clearly show that this is indeed background noise of the CRAC protocol?

2. Figure 4: 2'-O-methylation of the upper stem should lead to a stronger pairing and thus a more stable stem. To further test this, the first A-U pair (I guess 64) could be swapped to a G-C pair, which might overrule the strict requirement of a 2'-O-methylation at this position.

It is also interesting that the system seems to be not optimal since pairing of position 65 generates a much better substrate in vitro. What would be the evolutionary selection against such an optimal substrate? This could be further discussed.

3. The gradient shown in Fig. 5 are not fully clear. First, U4 is hardly visible and maybe a different probe could be used. U

snRNAs are also well stained by silver-staining protocols and since these are the major RNAs in nuclear lysates, this might be enough to visualize them.

The authors could perform the gradients under THUMP2 and LARP7 depletion conditions. This may enhance the effect and it may become clearer in these assays.

Reviewer #2

(Remarks to the Author)

In the manuscript by Bohnsack et al, the authors describe their systematic and structured approach to deciphering the installation of m2G in U6 snRNA and its biological consequences. I am impressed by the variety and at the same time high quality of the methods used by the whole team to answer important questions in the field of spliceosome assembly. The authors have used various and orthogonal methods and convincingly present data for the importance of m2G and its Nm interdependence in U6 snRNA. I see no need for additional experiments. From my perspective everything is covered. Yet, the Materials and Methods sections is not complete and requires careful reading and addition of information to ensure reproducibility.

- "Quantification of U6 snRNA modifications by LC-HRMS" is lacking detail. E.g. the digestion enzymes are not mentioned, column supplier is not mentioned, which timsTOF was used, flow rate of chromatography, the description of the gradient is unclear. The IP-MS data analysis section is more clear, but still misses some details, e.g. column information is missing, the injection volume, was a trap used?

Further, I have found the manuscript difficult to read and digest. Therefore, I suggest substantial editing of the main text and the discussion. I have made a collection of instances in which I struggled in hope that the manuscript can be advanced and more easy to parse.

- Some sentences require English editing. E.g. line 157 ff and line 471 ff.

- line 252: The current title is confusing and I suggest changing it to "Predicted RNA contacts localize to the THUMP and MTase domains" or something similar to emphasise the modelling character of the section

- line 376: The sentence "Bovine proteins were used as HsTHUMP2 expressed poorly" is unclear. If I understand correctly, the authors expressed the bovine orthologs of THUMP2 (and TRMT112?) because recombinant expression of the human protein was inefficient. This should be clarified explicitly to avoid misinterpretation.

- line 385 ff: The sentence spanning lines 384–389 is overly long and difficult to follow. I recommend breaking it into shorter sentences. The same is true for lines 252 ff, 276 ff, 333 ff, 422 ff and 511 ff.

- The discussion is more like a recap or summary. Interpretation is limited and missing or weakly developed. Method limitations (e.g. structural modeling and DNase sensitivity and its indirect character) are not discussed.

- N2,2-7-trimethylated cap (m3G): this nucleotide is commonly abbreviated as m227G according to modomics

I am confident that the work presented in this manuscript is a great fit for the broad readership of Nature Communications and I recommend publication after text revision.

Reviewer #3

(Remarks to the Author)

The manuscript by Kleiber et al. titled "2'-O-methylation-dependent installation of N2-methylguanosine in the U6 internal stem loop facilitates efficient spliceosome assembly" investigates how the m72G modification of the U6 snRNA is established by the THUMP2 enzyme and its reliance on the prior installation of 2'O methyl modifications on U6 snRNA by LARP7.

The manuscript has three major claims: 1- THUMP2 associates with U6 snRNA during late stages of its biogenesis; 2- m2G72 methylation requires 2'O methylation of U6 snRNA; and 3- m2G72 and 2'O methylations of U6 affect splicing interdependently.

Overall, the manuscript is well written with only minor writing errors and a few missing citations. The figures are well prepared and easy to understand. The materials and methods sections are detailed enough to reproduce the experiment. The manuscript makes a strong case for most of the claims and provides robust experimental data that clearly demonstrate the function of 2'O methylations in THUMP2 activity in vitro. It shows that the m72G modification is likely a late-stage modification in U6 biogenesis. The authors demonstrate that the absence of m72G likely affects spliceosome assembly and that the absence of THUMP2 and LARP7 impacts pre-mRNA splicing. The claims regarding the effects on pre-mRNA splicing could be supported more by additional analysis, as currently the effect of 2'O methylations is only tested in vitro. Nevertheless, overall, I believe this manuscript presents important data that will help the field better understand THUMP2 function and overall U6 snRNA biology.

Major comments

1- Line 149 – specify which human cells are over-expressing and note that they are over-expressing.

2- Line 149, I couldn't find any evidence or data showing this immunoprecipitations, its efficiency and its specificity in main or supplemental figures. It is important to demonstrate that protein and RNA immunoprecipitation are working effectively and specifically through western blots. Are the authors immunoprecipitating only a small percentage of the tagged protein or most of it? Similarly, how can they show that UV crosslinking is working effectively?

3- Similarly, Table S1 shows the overall reads mapped to different classes of RNAs or specific snRNAs, but it lacks information on the total reads (unmapped) from each experiment, the percentage mapped, read lengths, and so on.

4- Line 274, the main claim of the manual modelling is that it performs better than the AlphaFold-generated model. Therefore,

can the authors be more specific than “close proximity to THUMP2”? And following from this, why isn't G72 within the active site?

5- Figure 3i, similar to an earlier point about IP efficiency, makes it difficult to assess how effective their IP is since the input bands at 0.3% are stronger than the IP at ten times higher loading. The RNA binding data also suggest that THUMP2 binds only a very small fraction of U6 at any given time. Is that correct? Does the D329A mutant retain U6 longer?

6- Figure 4 provides a strong argument for the roles of 2'O methylation in G72 methylation by THUMP2. Additionally, in this figure, the authors demonstrate the use of a novel dnzyme-based method for detecting 2'O methylations. This is really good.

7- In Figure 5, authors make a good start to splicing analysis, but overall this section of the manuscript is weak. It was already known that the absence of THUMP2 and LARP7 affects splicing to some extent, and authors' data confirm this, showing there may be a stronger effect when both are missing. However, this does not tell us much about the specific role these modifications play in pre-mRNA splicing or why some genes are affected while others remain unaffected.

8- Figure 5i-k provides valuable insights into the mechanistic understanding of how these modifications influence spliceosome assembly. However, given its current format and the information provided, it is challenging to determine whether the conclusions are fully supported by the data. For example, in 5i, not only does U6 accumulate in fractions 10-17, but there is also a significant difference in U6 levels. Nevertheless, the authors show in figure s4 that the levels are the same. Additionally, there is little observable change in U4 or U5 levels or fractionation profiles. Furthermore, compared to Mabin et al., who performed similar fractionation, why is free U6 so low (PMID: 34234030)? Lastly, if less U6 is available for tri-snRNP assembly, why is the effect on splicing not more pronounced, impacting all transcripts and introns?

9- Can the authors show whether U6atac is affected in Figure 5i? The expectation is that U6atac should not be affected.

10- I don't like asking for additional complex experiments, but authors could consider testing whether spliceosome assembly, specifically of U6, is similarly affected in LARP7 KD cells compared to THUMP2. Currently, the 2'O methylation requirement for m7g is only evident from in vitro experiments. This could provide in vivo mechanistic insight into whether 2'O methylation can mimic THUMP2 absence. Alternatively, the wording around these results should more clearly indicate that the dependence is only evident in vitro.

11- I would like to see full-size gels of all Western and Northern blots as a supplementary file to ensure that the data is not selectively presented.

12- I appreciate the authors for providing a detailed materials and methods section.

Minor Comments

1- All prime symbols are incorrect. An apostrophe is not a prime symbol, and please ignore my use of one in my review report for ease of writing.

2- Line 118 – Citation for the METTL16 being the U6 snRNA m6A methyltransferase should include PMID: 28525753 and PMID: 29262316, and U6 snRNA A43 being involved in splicing should also cite PMID: 36409063 and PMID: 38808663.

3- Line 147 – Is a punctuation mark missing before “so”?

4- In Figure 2e, the GFP panel shows that the cell on the middle left has an incorrectly placed arrow.

Version 1:

Reviewer comments:

Reviewer #1

(Remarks to the Author)

The authors have adequately and competently responded to the few minor comments that I had raised on their previous version of the manuscript. Thus, I am satisfied with the response to my comments.

(Remarks on code availability)

Reviewer #2

(Remarks to the Author)

I am satisfied with the provided revision and I recommend the manuscript for publication.

(Remarks on code availability)

Reviewer #3

(Remarks to the Author)

Authors have made substantial revisions, including additional experiments that provide further support for their conclusions and additional information about the role of U6 2'O methylations.

They have addressed all the points I raised, and I appreciate the detailed, point-by-point response to our comments.

(Remarks on code availability)

We thank the referees for their positive feedback and constructive comments to our work. Below, we provide a complete point-by-point response to the reviewer's specific suggestions. The corresponding modifications to the text are indicated in red in the word files of the manuscript. In brief, we have added the following data to the manuscript:

Fig. 1a. Re-analysis of THUMPD2 CRAC datasets using peak-calling and feature overlap correction.

Fig. 1f. Extended western blot analysis of proteins co-immunoprecipitated with THUMPD2-His₆-2xFLAG with and without UV crosslinking.

Fig. 3i. Additional replicate northern blots to monitor association of wild-type and catalytic mutant THUMPD2 with the U6 snRNA.

Fig. 4o-p. *In vitro* methylation assays using THUMPD2-TRMT112 with stabilized U6 ISL substrate.

Fig. 5a-b. UpSet analysis of alternative splicing events in wild-type, THUMPD2 KO, LARP7 KD and THUMPD2 KO + LARP7 KD using new mRNA-seq data.

Fig. 5c-e. Sashimi plots showing examples of different types of alternative splicing events regulated by THUMPD2, LARP7 or both.

Fig. 5f. Distribution of significant and not significant inclusion differences for different AS types and conditions.

Fig. 6e. Gradient-based analysis of snRNA levels in snRNP complexes in cells lacking LARP7 or THUMPD2 and LARP7.

Fig. 6f. Quantification of snRNA levels in Fig. 6e.

Fig. 6g. Northern blot analysis of snRNA levels in nuclear extracts from wild-type, THUMPD2 KO, LARP7 KD and THUMPD2 KO + LARP7 KD cells.

Fig. 6i-l. Western blot analysis of PRPF3 (i), PRPF4 (j), PRPF8 (k) and EFTUD2 (l) migration profiles in gradients separating nuclear extracts from wild-type, THUMPD2 KO, LARP7 KD and THUMPD2 KO + LARP7 KD cells.

Supplementary Fig. S1a. Western blots showing recovery of THUMPD2-His₆-2xFLAG during immunoprecipitation.

Supplementary Fig. S1b. Autoradiograph showing RNA fragments crosslinked to THUMPD2-His₆-2xFLAG.

Supplementary Fig. S3f-g. *In vitro* methylation assays using THUMPD2-TRMT112 and and U6 ISL stabilized by nucleotide substitutions.

Supplementary Fig. S4a. Updated PCA analysis of new mRNA-seq data.

Supplementary Fig. S4b. Scheme of different types of alternative splicing.

Supplementary Fig. S4c. Updated heatmaps of significant alternative splicing events.

Supplementary Fig. S4d. Analysis of 5' and 3' splice site strength of retained introns.

Supplementary Fig. S4e. Cumulative IR fraction distribution of introns that change pre-mRNA splicing significantly.

Supplementary Fig. S6a. Analysis of U6atac in snRNPs in WT & THUMPD2 KO cells.

Supplementary Fig. S6b. U6atac levels in nuclear extracts from WT & THUMPD2 KO cells.

Supplementary Table S1. Output values of CRAC sequencing read alignments.

Reviewer 1

Reviewer: “In this manuscript, the authors have biochemically and functionally characterized the enzyme complex that m²G-modifies the U6 snRNA at position 72. Although this link has been made before, the authors show that this is the only modification that THUMPD2 generates. THUMPD2 is enriched in Cajal bodies, it associates with the adaptor protein TRMT112 and binds to U6 at later stages of U6 RNP assembly. The authors further use crosslinking experiments to map the interaction of THUMPD2 to U6 and TRMT112. U6 is contacted at multiple regions in the N and the C terminal part of the protein, particularly the THUMP and the MTase domain. TRMT112 was mapped to the C terminus of the protein. They further generated a DNAzyme that allows for the site-specific identification of m²G72. THUMPD2 knock out cells clearly show reduced modification at this position and complementation assays revealed that SAM binding and TRMT112 binding is required for methylation in cells. However, in vitro, the reaction was not efficient and therefore the authors investigated the contribution of associated 2'-O-methylation events. Indeed, pre-2'-O-methylated substrates were efficiently modified by the reconstituted enzyme complex in vitro. Using LARP7 knock down, which globally reduced U6 2'-O-methylation, the authors further show that prior U6 2'-O-methylation is also required in cells. Finally, the authors find that alternative splicing events are affected by THUMPD2 knock out, LARP7 knock down and a stronger effect when both are inactivated simultaneously. This appears to be due to less efficient di- and tri-snRNP formation.

This is a very solid and clear manuscript. All experiments are competently carried out, presented and interpreted. Although the link between THUMPD2 and U6 m²G72 has already been made, the manuscript goes far beyond and presents a thorough characterization of the entire pathway including experiments in cells, generation of new tools such as DNAzymes as well as biochemical in vitro reconstitution of the enzymatic activities. I have only a few comments that could be considered.”

Reply: We thank the reviewer for the positive feedback on our work and manuscript.

Reviewer: “1. Fig. 1A: about 60% of the reads are crosslinked to lncRNAs or mRNAs (which is also seen in the negative control). It is clear that no modification sites besides the one in the ISL of U6 were found. However, THUMP2 may bind to other transcripts without methylating them. Have some of the mRNAs or lncRNAs been validated to clearly show that this is indeed background noise of the CRAC protocol?”

Reply: Upon closer inspection of the distribution of sequencing reads mapping to protein-coding genes in the THUMP2 CRAC datasets, we observed that most mapped to introns that contain U6 sequences. This overlap is illustrated by the example genome coverage traces (IGV snapshots) for three mRNAs with high numbers of reads (>800 reads per gene;

Reply Fig. 1). In the previous bioinformatics analysis, which monitored the normalized (RPM) numbers of reads mapping to different genes, reads mapping to these genes were attributed to mRNAs, whereas they actually reflect reads derived from U6.

To correct this mis-assignment, we re-analysed the CRAC datasets using peak-calling, and only reads mapping to genomic regions (peaks) significantly enriched in both the THUMP2 datasets compared to the controls were used for feature count analysis considering “exons” rather than “genes”. For both datasets, this demonstrated that the vast majority of sequencing reads in the THUMP2 dataset were derived from the U6 snRNA. A degree of variability between the two datasets was observed (80% of the mapped reads significant after peak calling were mapping to U6 in replicate 1 and 61% in replicate 2), likely reflecting technical differences during the library preparation, so instead of averaging the data, the two datasets are rather presented independently in **Fig. 1a** and **Source Data Fig. 1a**.

Reply Figure 1. Mapping profiles of THUMP2-His₆-2xFLAG and His₆-2xFLAG CRAC data on selected genes. **(a)** Reads mapping to the *CDK6* intron overlapping with *RNU6-10P*. **(b)** Reads mapping to the *GRM5* intron overlapping with *RNU6-16P*. **(c)** Reads mapping to the *KIF1B* intron overlapping with *RNU6-37P*.

Reviewer: “2. Figure 4: 2'-O-methylation of the upper stem should lead to a stronger pairing and thus a more stable stem. To further test this, the first A-U pair (I guess 64) could be swapped to a G-C pair, which might overrule the strict requirement of a 2'-O-methylation at this position. It is also interesting that the system seems to be not optimal since pairing of position 65 generates a much better substrate *in vitro*. What would be the evolutionary selection against such an optimal substrate? This could be further discussed.”

Reply: *In vitro* methylation assays were performed on an RNA substrate in which the U₆₄-A₇₀ pair at the top of the U6 ISL was exchanged for a C₆₄-G₇₀ pair. This oligonucleotide was not methylated by THUMPD2-TRMT112 under the conditions used (**Reply Fig. 2**). However, similar to the RNA containing the endogenous U₆₄-A₇₀ basepair, addition of 2'-O-methyl groups to C₆₂ and C₆₃ to the substrate containing the C₆₄-G₇₀ pair was sufficient to stimulate THUMPD2 activity *in vitro* (**Fig. 4f** and **Reply Fig. 2**). To further explore the influence of the stability of the U6 ISL on m²G₇₂ installation by THUMPD2-TRMT112, *in vitro* methylation assays were performed using a version of the U6 ISL containing four nucleotide substitutions stabilizing the stem (C₆₁U, G₆₅U, C₆₆G and A₇₆C). These changes create two A-U and two G-C basepairs where there are non-conventional basepairs in the wild-type sequence. This oligonucleotide was also not a substrate of THUMPD2-TRMT112 *in vitro* (**Fig. 1**).

Reply Figure 2. *In vitro* methylation assay with THUMPD2-TRMT112 and U6 ISL substrates containing nucleotide exchanges and modified/unmodified nucleotides. (a-c) Schematic views of the RNA substrates used for *in vitro* methylation assays. **(d)** *In vitro* methylation assays were performed using purified His₆-ZZ-B τ THUMPD2-B τ TRMT112 protein complex, [³H]-SAM as methyl group donor and the specific RNA substrates indicated in (a-c). Tritium incorporated in the substrate RNAs was quantified and bar plots show averaged counts per minute (CPM) of n=3 independent experiments with error bars representing the standard deviation. Significance was calculated using unpaired t-tests. *p < 0.05 and ns = not significant.

We conclude, therefore, that the stability of the stem does not overcome the requirement for the Cm₆₂ and Cm₆₃ 2'-O-methylations, but that certain mutations stabilizing the upper part of the stem increase *in vitro* methylation of U6-G₇₂ in the presence of Cm₆₂ and Cm₆₃. The *in vitro* methylation data for the maximally stabilized U6 ISL are presented in the revised manuscript in **Fig. 4p**.

The evolutionary selection against a U6 ISL with a more stable stem is likely a result of the necessity for U6 to assume multiple conformations during its assembly and integration into spliceosomes. In the U6 snRNP, nucleotides 55-80 fold into an intramolecular stem-loop, whereas in the U4-U6 di-snRNP, this helix is completely unwound to basepair with U4 and, in the active spliceosome, this sequence folds back into an intramolecular helix when U6 is base-paired with U2.

Mutational analysis of U6 transcripts used in experiments with human nuclear extracts show that some nucleotide substitutions that (over-)stabilize the U6 ISL can be detrimental for U4-U6 interaction, snRNP assembly and splicing. For instance, A₇₃G substitution in the upper part of the U6 ISL stabilizes the C₆₁-A₇₃ pair immediately upstream of the bulged U₇₄ and was shown to reduce U4-U6 interaction and spliceosome assembly (PMID: 8330741). Similar experiments also reveal that A₇₆C, which stabilizes the G₅₉-A₇₆ pair close to the catalytic U₇₄ leads to loss of most of the U6-U4 interaction *in vitro*, significantly reducing spliceosome assembly and splicing activity (PMID: 7632731). Additionally, an A₅₆G substitution that stabilizes the A₅₆-C₇₉ pair in the lower stem region of the U6 ISL also reduces U6-U4 interaction (PMID: 7632731). In yeast, it has also been shown that stabilizing the upper loop of the U6 ISL via substitution of C₇₂ (equivalent to C₆₆ in the human U6 ISL) for A₇₂ contributes to a strong cold-sensitive phenotype, likely arising due to perturbed pre-mRNA splicing (PMID: 31229405). Data from yeast also suggests that hyperstabilizing U6 ISL structures can interfere with the catalytic conformation of the U6 ISL (PMID: 31229405). For example, a C₆₇A substitution that allows base-pairing with U₈₀ (equivalent to U₇₄ in the human U6 ISL) is proposed to inhibit docking of U₈₀ into the catalytic triplex, stabilizing a conformation of the ISL that is not properly functional (PMID: 31229405). Additionally, cells expressing an A₇₉G mutant that stabilizes the C₆₇-A₇₉ pair in close proximity to the catalytic site exhibit a cold-sensitive phenotype, suggesting that formation of a rigid structure around the catalytic site can hinder interactions of U₈₀ with the catalytic triplex (PMID: 31229405).

Furthermore, it has been shown that, across different species, the upper ISL segment contains several conserved C-G pairs, while the lower ISL segment has a higher sequence variability, containing frequent weak and non-canonical base-pairings that confer higher ISL flexibility (PMID: 31229405). The conservation of these features suggests that they ensure an optimal balance between stability and flexibility that allows the required rearrangements of this region during the transitions between the different steps in snRNP assembly and

splicing. This topic is now addressed more extensively in the discussion section of the manuscript.

Reviewer: “3. The gradient shown in Fig. 5 are not fully clear. First, U4 is hardly visible and maybe a different probe could be used. U snRNAs are also well stained by silver-staining protocols and since these are the major RNAs in nuclear lysates, this might be enough to visualize them. The authors could perform the gradients under THUMPD2 and LARP7 depletion conditions. This may enhance the effect and it may become clearer in these assays.”

Reply: The previously poor detection of U4 relative to the other snRNAs did indeed hamper interpretation of the gradients, so the northern blot probe used to detect U4 was changed and the amounts of each probe used were titrated so all five snRNAs can be visualised well and with similar signal intensity. **Reply Fig. 3** shows the difference before and after optimizing snRNA detection in the gradients by northern blotting. All northern blots presented in the updated version of **Fig. 6** (and the replicates shown in **Source Data Fig. 6e**) have been re-probed using the optimized approach.

Reply Figure 3. Optimization of northern blotting to detect snRNAs. Representative northern blots show the detection of snRNAs of the major spliceosome using the previous strategy (upper), and with an alternative U4 probe and titrated levels of each probe (lower) in the same membrane.

The gradient-based analysis of the effect of THUMPD2 absence on snRNP complex assembly and integration into the spliceosome has been extended to include the depletion of LARP7 and the combination of LARP7 KD in the THUMPD2 KO cell line. It is important to note that due to the requirement of prior 2'-O-methylation of C₆₂ and C₆₃ for efficient THUMPD2-TRMT112-mediated methylation (**Fig. 4h-i**), in both these conditions, m²G₇₂ is reduced. Interestingly however, we observe that lack of m²G₇₂ alone and the reduction of 2'-O-methylations together with reduced/absent m²G₇₂ have different effects on U6 snRNA levels in snRNPs. Lack of m²G₇₂ alone leads to accumulation of the U6 snRNA in early

gradient fractions, indicating this modification is important for the progression of U6-containing di-/tri-snRNP complexes into larger spliceosomal complexes, whereas upon depletion of LARP7, a strong reduction in U6 incorporation into snRNPs is observed. This suggests that the U6 2'-O-methylations are functionally important prior to the m²G₇₂, and that phenotypically, the lack of U6 2'-O-methylations caused by LARP7 KD therefore predominates over the absence of m²G in the THUMPD2 KO.

To further consolidate our conclusions about the effect of lack of m²G on spliceosomal complex assembly, we have also probed our gradients for additional di-snRNP and tri-snRNP markers. Both PRPF4 and PRPF3 show marked differences in their migration profiles between WT and THUMPD2 KO, supporting sub-optimal progression of U6 snRNA-containing snRNPs into spliceosomal complexes due to lack of m²G₇₂.

These new data are presented in an updated **Fig. 6** and the effects of the different modifications are discussed in the revised manuscript.

Reviewer 2

Reviewer: “In the manuscript by Bohnsack et al, the authors describe their systematic and structured approach to deciphering the installation of m²G in U6 snRNA and its biological consequences. I am impressed by the variety and at the same time high quality of the methods used by the whole team to answer important questions in the field of spliceosome assembly. The authors have used various and orthogonal methods and convincingly present data for the importance of m²G and its Nm interdependence in U6 snRNA. I see no need for additional experiments. Form my perspective everything is covered. Yet, the Materials and Methods sections is not complete and requires careful reading and addition of information to ensure reproducibility.”

Reply: We thank the reviewer for the positive feedback on our work and the suggestions to improve the manuscript.

Reviewer: “Quantification of U6 snRNA modifications by LC-HRMS” is lacking detail. E.g. the digestion enzymes are not mentioned, columns supplier is not mentioned, which timsTOF was used, flow rate of chromatography, the description of the gradient is unclear. The IP-MS data analysis section is more clear, but still misses some details, e.g. column information is missing, the injection volume, was a trap used?”

Reply: We have added the missing details to the LC-HRMS and IP-MS method sections. Please note that the timsTOF used is a first generation machine, so is only named “timsTOF”, without the additional differentiations of next generation machines.

Reviewer: “Further, I have found the manuscript difficult to read and digest. Therefore, I suggest substantial editing of the main text and the discussion. I have made a collection of instances in which I struggled in hope that the manuscript can be advanced and more easy to parse.”

Reply: We thank the reviewer for highlighting instances where sentences can be improved and have incorporated the suggestions pointed out below. In the light of this comment and to conform to editorial length requirements, the text has been substantially edited. Changes introduced for text compaction, which do not alter the meaning or conclusions, are not highlighted in red.

Reviewer: “Some sentences require English editing. E.g. line 157 ff and line 471 ff.”

Reply: The sentences on lines 157 and 471 have been removed in the context of describing the updated analysis of the CRAC data and the new *in vitro* methylation data.

Reviewer: “line 252: The current title is confusing and I suggest changing it to “Predicted RNA contacts localize to the THUMP and MTase domains” or something similar to emphasise the modelling character of the section”

Reply: The RNA contact sites in THUMPD2 were detected experimentally using UV crosslinking coupled to mass spectrometry to identify amino acids physically associated with RNA nucleotides. As the U6 snRNA is likely the sole target of THUMPD2 (**Fig. 1a**) and UV crosslinking is considered a “zero-distance” approach, the crosslinking sites identified can be considered as relevant RNA contact sites of THUMPD2 with U6. The sub-heading has been updated to “RNA contact sites on THUMPD2 are present in the THUMP and MTase domains”. The modelling approach used to generate the structure on which these contact sites are depicted in **Fig. 3c** is described in detail in the text.

Reviewer: “line 376: The sentence “Bovine proteins were used as HsTHUMPD2 expressed poorly” is unclear. If I understand correctly, the authors expressed the bovine orthologs of THUMPD2 (and TRMT112?) because recombinant expression of the human protein was inefficient. This should be clarified explicitly to avoid misinterpretation.”

Reply: It is indeed the case that human THUMPD2-TRMT112 expressed poorly in *E. coli* and for this reason, the bovine proteins were used instead. This has now been clarified in the text.

Reviewer: “line 385 ff: The sentence spanning lines 384–389 is overly long and difficult to follow. I recommend breaking it into shorter sentences. The same is true for lines 252 ff, 276 ff, 333 ff, 422 ff and 511 ff.”

Reply: The highlighted sentences have been amended to aid readability.

Reviewer: “The discussion is more like a recap or summary. Interpretation is limited and missing or weakly developed. Method limitations (e.g. structural modeling and DNazyme sensitivity and its indirect character) are not discussed.”

Reply: The discussion section has been revised to describe the methodological approaches in more detail and include further interpretation of the data.

Reviewer: “N2,2-7-trimethylated cap (m3G): this nucleotide is commonly abbreviated as m227G according to modomics”

Reply: The abbreviation used in the Modomics database has been added.

Reviewer: “I am confident that the work presented in this manuscript is a great fit for the broad readership of Nature Communications and I recommend publication after text revision.”

Reply: We thank the reviewer for recommending our work for publication in Nature Communications.

Reviewer 3

Reviewer: “The manuscript by Kleiber et al. titled “2'-O-methylation-dependent installation of N2-methylguanosine in the U6 internal stem loop facilitates efficient spliceosome assembly” investigates how the m72G modification of the U6 snRNA is established by the THUMP2 enzyme and its reliance on the prior installation of 2'O methyl modifications on U6 snRNA by LARP7.

The manuscript has three major claims: 1- THUMP2 associates with U6 snRNA during late stages of its biogenesis; 2- m2G72 methylation requires 2'O methylation of U6 snRNA; and 3- m2G72 and 2'O methylations of U6 affect splicing interdependently. Overall, the manuscript is well written with only minor writing errors and a few missing citations. The figures are well prepared and easy to understand. The materials and methods sections are detailed enough to reproduce the experiment. The manuscript makes a strong case for most of the claims and provides robust experimental data that clearly demonstrate the function of 2'O methylations in THUMP2 activity in vitro. It shows that the m72G modification is likely a late-stage modification in U6 biogenesis. The authors demonstrate that the absence of m72G likely affects spliceosome assembly and that the absence of THUMP2 and LARP7 impacts pre-mRNA splicing. The claims regarding the effects on pre-mRNA splicing could be supported more by additional analysis, as currently the effect of 2'O methylations is only tested in vitro. Nevertheless, overall, I believe this manuscript presents

important data that will help the field better understand THUMPD2 function and overall U6 snRNA biology.”

Reply: We thank the reviewer for the positive feedback on our work and manuscript.

Major comments

Reviewer: “1- Line 149 – specify which human cells are over-expressing and note that they are over-expressing.”

Reply: The cell type used is now mentioned in the Results section as well as the fact that THUMPD2-His₆-2xFLAG is over-expressed.

Reviewer: “2- Line 149, I couldn’t find any evidence or data showing this immunoprecipitations, its efficiency and its specificity in main or supplemental figures. It is important to demonstrate that protein and RNA immunoprecipitation are working effectively and specifically through western blots. Are the authors immunoprecipitating only a small percentage of the tagged protein or most of it? Similarly, how can they show that UV crosslinking is working effectively?”

Reply: A western blot showing the efficiency of the anti-FLAG immunoprecipitation of THUMPD2-His₆-2xFLAG has been included as **Supplementary Fig. S1a**. After quantification of the inputs and eluates in n=3 experiments, an average immunoprecipitation efficiency of 37% was obtained for wild-type THUMPD2-His₆-2xFLAG. Further supporting an adequate recovery of THUMPD2-His₆-2xFLAG in the immunoprecipitation experiments, very little FLAG-tagged protein was left in the flow-through (**Reply Fig. 4**). Uncropped western blots for immunoprecipitations are now provided in **Source Data Fig. 1f** and **Fig. 3i**.

Reply Figure 4. Immunoprecipitation of THUMPD2-His₆-2xFLAG. Input, flow-through and eluate samples from an anti-FLAG immunoprecipitation experiment performed as in Fig. 1f and Fig. 3i were analyzed by western blotting using the indicated antibodies.

It is known that the RNA–protein crosslinking efficiency using UV is naturally very low (PMID: 35581222). Nevertheless, the strong enrichment of the U6 snRNA with THUMPD2-His₆-2xFLAG in CRAC and RNA-IP experiments (**Fig. 1a**, **Supplementary Fig. S1b** and **Fig. 3i**),

which include a purification step under highly denaturing conditions (6 M guanidine hydrochloride), demonstrates the formation of crosslinked RNA–protein complexes.

The effectiveness of UV crosslinking in stabilizing the THUMP2–U6 snRNA interaction is demonstrated by comparing the amounts of U6 recovered with THUMP2-His₆-2xFLAG in immunoprecipitation experiments performed with and without prior UV crosslinking of cells. In the IP performed without prior crosslinking (native), low amounts of U6 snRNA were recovered whereas after UV crosslinking, the U6 signal obtained by northern blotting was markedly higher (**Reply Fig. 5**). Furthermore, the U6 signal detected in the IP eluate from the crosslinked cells is shifted upwards relative to U6 recovered from the non-crosslinked cells, which is consistent with the formation of covalently crosslinked complexes. These data are now presented in **Supplementary Fig. S1c** of the revised manuscript.

Reply Figure 5. Immunoprecipitation of THUMP2-His₆-2xFLAG from UV crosslinked and non-crosslinked cells. Immunoprecipitation experiment with THUMP2-His₆-2xFLAG (performed as in Fig 1f), showing recovery of the U6 snRNA without (native) or with UV crosslinking (UV) by northern blotting. In (b) the uncropped Northern blot for (a) is shown.

The stabilizing effect of UV crosslinking on THUMP2-containing complexes is further emphasized by the comparison of proteins recovered in immunoprecipitation experiments with and without UV crosslinking as shown in **Fig. 1f**. IP eluates analyzed in parallel on the same membranes (**Reply Fig. 6**, right panel) show that although TRMT112 and FBL are recovered similarly in both IPs, EFTUD2 and MEPCE are more efficiently recovered after UV crosslinking.

Reply Figure 6. Comparison of proteins recovered with THUMPDP2-His₆-2xFLAG in IP experiments with and without prior UV crosslinking. Extracts from cells expressing His₆-2xFLAG or THUMPDP2-His₆-2xFLAG that were left untreated or UV crosslinked were used for anti-FLAG immunoprecipitation experiments. Inputs (left panel) and eluates (right panel) were analyzed by western blotting using the indicated antibodies.

Reviewer: “3- Similarly, Table S1 shows the overall reads mapped to different classes of RNAs or specific snRNAs, but it lacks information on the total reads (unmapped) from each experiment, the percentage mapped, read lengths, and so on.”

Reply: Parameters including number of input reads, average read length, percentage of uniquely mapped reads are now presented in **Supplementary Table S1**.

Reviewer: “4- Line 274, the main claim of the manual modelling is that it performs better than the AlphaFold-generated model. Therefore, can the authors be more specific than “close proximity to THUMPDP2”? And following from this, why isn’t G72 within the active site?”

Reply: As initially indicated in the Material and Methods section, this manual model was obtained by superposing AlphaFold models of the THUMPDP2-TRMT112 complex, previously determined structures of the U6 snRNA ISL and a complex between a THUMP domain bound to a double stranded RNA. It can be considered, therefore, as a rigid body model. In this model, G₇₂ is facing the THUMPDP2 active site but is engaged in a Watson-Crick base pair with C₆₂ of the U6 ISL. A simple flipping-out mechanism, commonly observed in structures of double-stranded RNAs (in particular tRNAs and rRNAs) bound to RNA modifying enzymes, could project the G₇₂ base into the THUMPDP2 active site, in an ideal position for N2 of G₇₂ to be opposite the SAM methyl group to be transferred. It was decided not to model this flipped-out G₇₂ base into the THUMPDP2 active site to limit over-modeling. However, the proposed mechanism is described in the revised manuscript.

Reviewer: “5- Figure 3i, similar to an earlier point about IP efficiency, makes it difficult to assess how effective their IP is since the input bands at 0.3% are stronger than the IP at ten times higher loading. The RNA binding data also suggest that THUMPDP2 binds only a very small fraction of U6 at any given time. Is that correct? Does the D329A mutant retain U6 longer?”

Reply: Uncropped western blots of inputs and eluate samples from immunoprecipitations show the relative enrichment of wild-type THUMPDP2-His₆-2xFLAG and the mutant versions and allow the IP efficiency to be determined (**Fig. 3i**, **Source Data Fig. 3i** and **Reply Fig. 7**).

Reply Figure 7. Immunoprecipitation of wild-type THUMPDP2-His₆-2xFLAG and mutant versions. The relative expression levels of wild-type THUMPDP2-His₆-2xFLAG and its derivatives in cells are shown (input) as well as their recovery after anti-FLAG immunoprecipitation (eluate). Protein samples were analyzed by western blotting using an anti-FLAG antibody.

The IP efficiency (%) was calculated in n=3 experiments by quantifying eluate and input bands in the relevant blots (**Reply Table 1**). In all cases, at least a third of the bait protein was recovered in the eluates.

Reply Table 1. Quantification of IP efficiency of wild-type THUMPDP2-His₆-2xFLAG and its derivatives. Values reflect (eluate*0.3)/(input*3.3)*100.

	Rep. #1	Rep. #2	Rep. #3	Average
THUMPDP2	47.0	33.5	32.4	37.6
D329A	40.2	34.9	43.3	39.5
E352R	33.4	32.6	30.8	32.2
NTD	85.0	92.2	33.9	70.4
CTD	33.1	16.3	41.7	30.4

It is indeed the case that only a small proportion of the U6 snRNA is recovered with THUMPDP2-His₆-2x-FLAG. The U6 snRNA has an estimated half-life of 24 h (PMID: 8892983) and our data suggest that THUMPDP2 associates with nascent U6 during its maturation in the nucleoplasm/Cajal bodies (**Fig. 2**), likely dissociating prior to incorporation of the U6 snRNP into di-snRNP complexes. The transient nature of the interaction of THUMPDP2 with the U6 snRNAs is supported by the finding that TRMT112 is the only robust interaction partner of THUMPDP2 identified by IP-MS after crosslinking (**Fig. 1e**). Most of the RNA detected in the

input samples probably represents U6 snRNA incorporated into snRNPs and other spliceosomal complexes whereas only a small proportion of newly synthesized U6 undergoing maturation is captured in association with THUMPD2. The low recovery of U6 snRNA with THUMPD2 is rationalized in the revised manuscript.

Considering the relative recovery of U6 with wild-type THUMPD2 and the catalytic mutant (D329A), it is not possible to accurately quantify the amount of U6 co-precipitated relative to the bait protein as these samples require to be analyzed by northern and western blotting, respectively (i.e. separate gels/membranes). To allow better estimates of the relative enrichment of U6, this experiment has now been performed n=5 times. Considering all data, it appears that slightly higher amounts of U6 are usually recovered with the THUMPD2 mutant compared to the wild-type protein. The data from all experiments are shown in **Source Data Fig. 3i**, and another representative blot (in addition to **Fig. 3i**) is shown below (**Reply Fig. 7**). The relevant section of the text has been amended to reflect this observation.

Reply Figure 7. RNA-IP of U6 with wild-type and catalytically inactive THUMPD2. Extracts from UV crosslinked cells over-expressing His₆-2xFLAG, THUMPD2-His₆-2xFLAG and THUMPD2(D329A)-His₆-2xFLAG were used in anti-FLAG immunoprecipitation experiments. Proteins in inputs (left panel) and eluates (right panel) were analyzed by western blotting using the indicated antibodies (upper), and U6 was detected by northern blotting (lower).

Reviewer: “6- Figure 4 provides a strong argument for the roles of 2’O methylation in G72 methylation by THUMPD2. Additionally, in this figure, the authors demonstrate the use of a novel dnzyme-based method for detecting 2’O methylations. This is really good.”

Reply: We thank the reviewer for appreciating the value of the developed DNazymes to detect m²G and 2’-O-methylations in a site-specific manner.

Reviewer: “7- In Figure 5, authors make a good start to splicing analysis, but overall this section of the manuscript is weak. It was already known that the absence of THUMPD2 and

LARP7 affects splicing to some extent, and authors' data confirm this, showing there may be a stronger effect when both are missing. However, this does not tell us much about the specific role these modifications play in pre-mRNA splicing or why some genes are affected while others remain unaffected.”

Reply: In the initial submission, the analyses of alternative splicing in cells lacking THUMPD2 and/or LARP7 were performed on total RNA-seq datasets (with rRNA depletion). To improve our comparative alternative splicing analyses, poly(A) enrichment and mRNA-seq was now performed to enhance the depth of the sequencing. This increased the number of significant alternative splicing events detected in all conditions, thereby allowing to consolidate the conclusions and probe further features of these datasets. Consistent with the observations that particular 2'-O-methylations in the ISL promote installation of m²G₇₂ (**Fig. 4**), but that lack of U6 2'-O-methylations and m²G₇₂ differently influence snRNP assembly (**Fig. 6**), we detect numerous alternative splicing events that are independently and interdependently affected by lack of THUMPD2, LARP7 or both (**Fig. 5; Supplemental Fig. S4**). The scope of our analysis of alternative splicing extends to include: i) identification of different types of alternative splicing events in all datasets and comparative analysis of those significant in different conditions with an improved visualization using UpSet plots, ii) highlighting three examples, representing the intron retention upon THUMPD2 KO, alternative 3' splice site variation upon LARP7 KD and a skipped exon event that is significantly different from WT, iii) narrowing down types of AS that are specifically affected either upon THUMPD2 KO (intron retention) or LARP7 KD (A3SS), and iv) addressing the combinatorial nature of the two perturbations. Related to i), we modified the analysis for intron retention to include more introns beyond what is possible with rMATS. We quantify intron retention using split reads and exon-intron junction reads for 37885 introns that pass our read count cutoff in all conditions. As expected most introns are well spliced. This is visualized in the new **Supplemental Fig. S4e**. Introns that show a significant increase or decrease in splicing for any of the perturbations tend to be already less efficiently spliced (**Fig. 5c; Supplemental Fig. S4e**). For intron retention, we observe that THUMPD2 KO decreases splicing-efficiency. To visualize all types of AS, we moved away from using Venn diagrams to UpSet plots that better capture the intersections. A3SS, A5SS, MXE and SE are grouped together as they all originate from the AS analysis using rMATS.

We modified the text, figures, tables, as well as the GEO submission to reflect these adjustments.

With these analyses, we extend our previous conclusions, which mainly focused on the increase in significant events for the combined perturbation, by highlighting the importance of LARP7/U6 2'-O-methylation for 3' splice site selection, which is distinct from THUMPD2/m²G₇₂ that affects intron retention in a directed manner. A modest correlation of

5' and 3' SS scores with intron retention was detected (**Supplemental Fig. S4d**), but no sequence features associated with the increase in distal A3SS usage upon LARP7 KD were observed. Nevertheless, the identified distinction between the effects of the different types of modifications is a good basis for further analyses in the future, e.g. with regard to possibly altered co-transcriptional aspects of spliceosome assembly.

Reviewer: “8- Figure 5i-k provides valuable insights into the mechanistic understanding of how these modifications influence spliceosome assembly. However, given its current format and the information provided, it is challenging to determine whether the conclusions are fully supported by the data. For example, in 5i, not only does U6 accumulate in fractions 10-17, but there is also a significant difference in U6 levels. Nevertheless, the authors show in figure s4 that the levels are the same. Additionally, there is little observable change in U4 or U5 levels or fractionation profiles. Furthermore, compared to Mabin et al., who performed similar fractionation, why is free U6 so low (PMID: 34234030)? Lastly, if less U6 is available for tri-snRNP assembly, why is the effect on splicing not more pronounced, impacting all transcripts and introns?”

Reply: The 10-30% glycerol gradients performed under the conditions mentioned in the manuscript (34,000 rpm for 18 h in a SW40 rotor, using 100 mM KCl) allow resolution of snRNPs such that the 12S (U1 snRNP), 17S (U2 snRNP) and 25S (U6:U4.U5 snRNP) complexes are very well separated. However, these gradients do not provide information on the levels of the snRNAs in the spliceosomal complexes (A-complex, B-complex, etc.) as this would require very different conditions. While the overall levels of U6 are not affected by THUMPD2 KO (**Fig. 6g**), the increased U6 levels in fractions 10-17 represent an accumulation of U6 in the complexes migrating in these fractions. Accumulation of U6 in the mono-/di-/tri-snRNP complexes present in fractions 10-17 in the THUMPD2 KO indicates that less U6 progresses into later spliceosomal complexes. Although the accumulation of U6 in these fractions in the THUMPD2 KO is more pronounced than for U4 and U5, the levels of these snRNAs in these fractions also increase. After improving the detection of U4 snRNA by exchanging the northern blotting probe, the accumulation of this snRNA becomes more apparent (**Fig. 6e-f**). Interestingly, we observed that depletion of LARP7 (see point 10) similarly has a stronger effect on the amount of U6 present in these complexes than U4 or U5 (**Fig. 6e-f**). The description of these results has been improved to clarify the conclusions drawn from the blots.

In the study by Mabin et al., the gradients were performed using extracts from 293T cells and more free U6 (see Figure S7D of PMID: 34234030; fractions 5-7) was detected than in our gradients, which were performed with nuclear extracts from HCT116 cells. It is possible that the differences in levels of free U6 arise due to variations between the cell lines. For example,

in Tanackovic et al., similar gradients performed using extracts from lymphoblasts (see Figure 2 of PMID: 21378395) show low amounts of free U6. However, we can not exclude the possibility that lesser U6 incorporated into di-/tri-snRNPs could be explained by differences in the preparation of the extracts or gradients. For comparison, in Yildirim et al., glycerol gradients also performed with extracts from HEK293T cells (see Figure S5A of PMID: 34023904; fractions 5-7) show less free U6, similar to the results we obtained.

As described above, our analyses of alternative splicing in THUMPD2 KO cells in the first submission were based on total RNA-seq (ribosomal RNA depletion). Upon increasing the sequencing depth by performing mRNA-seq (polyA-enrichment), we observe many more alternative splicing events that are significantly affected by loss of THUMPD2 (**Fig. 5**). The accumulation of U6 in di-/tri-snRNP complexes in the absence of m²G₇₂ (**Fig. 6**) suggests that this modification optimizes the maturation of these complexes and their integration into larger splicing assemblies. The mild effects on splicing observed in the THUMPD2 KO are consistent with perturbation of spliceosome assembly kinetics rather than a block in tri-snRNP formation. By contrast, our new data show that LARP7 depletion strongly impairs U6 incorporation into di-/tri-snRNP complexes (see also point 10), and consistent with this, the effects of LARP7 KD on splicing are more pronounced than those arising from THUMPD2 KO. Notably, there are many more alternative splicing events detected in the THUMPD2 KO, but to increase confidence in the reported events, we chose to present only the most robust changes. However, the detection of numerous mild effects on alternative splicing that are below the stringent significance threshold is in line with the proposed, non-essential role of m²G₇₂ in fine-tuning spliceosome function. It is also important to consider the possibility of adaptation in the knockout cell line that may partially compensate for the lack of THUMPD2 and thus contribute to the relatively low number of significant alternative splicing (AS) events in the KO cell line compared to wild-type. It is also worth mentioning that this type of analysis does not provide information on the efficiency of splicing catalysis, i.e. if splicing is slower/faster than normally. We hypothesise that disturbance of snRNP assembly caused by lack of U6-m²G₇₂, may reduce overall splicing efficiency, which could explain the growth defect observed in THUMPD2 KO cells (PMID: 37283053). These topics are now discussed in more detail in the revised manuscript.

Reviewer: “9- Can the authors show whether U6atac is affected in Figure 5i? The expectation is that U6atac should not be affected.”

Reply: Northern blot analysis of U6atac in nuclear extracts shows that its overall level is not affected by THUMPD2 KO (**Supplementary Fig. S6b**). Reprobing of the gradient membranes to detect U6atac, together with U1 snRNA as reference, suggests a minimal effect of THUMPD2 KO on the level of U6atac in mono-/di-/tri-snRNP complexes when

compared to wild-type (**Supplementary Fig. S6a**). Our data show that lack of THUMPD2 affects the migration profiles of PRPF3 and PRPF4 in these gradients (**Fig. 6i-j**); as it has previously been observed that mutations in PRPF proteins, which associate with both U6 and U6atac, affect levels of both snRNAs incorporated into snRNPs (PMID: 21378395), we speculate that the mild effects of THUMPD2 KO on U6atac may arise due the impact of U6- m^2G_{72} on PRPF3/4 association with snRNP particles. These new data are presented in the revised manuscript as **Supplementary Fig. S6**.

Reviewer: “10- I don’t like asking for additional complex experiments, but authors could consider testing whether spliceosome assembly, specifically of U6, is similarly affected in LARP7 KD cells compared to THUMPD2. Currently, the 2’O methylation requirement for m7g is only evident from in vitro experiments. This could provide in vivo mechanistic insight into whether 2’O methylation can mimic THUMPD2 absence. Alternatively, the wording around these results should more clearly indicate that the dependence is only evident in vitro.”

Reply: The requirement for 2’-O-methylations in U6 for THUMPD2-TRMT112-mediated installation of m^2G_{72} was demonstrated both *in vitro* (**Fig. 4 e-f**) and in cellular RNAs. In **Fig. 4g-i**, primer extension (**Fig. 4h**) and DNAzyme cleavage assays (**Fig. 4i**, upper and middle panels) confirm a reduction in 2’-O-methylations in U6 upon LARP7 KD. In parallel, we show by DNAzyme cleavage (**Fig 4i**, bottom panel) that the levels of m^2G_{72} are also affected.

We have performed gradients with nuclear extracts prepared from cells depleted of LARP7 in either the wild-type or THUMPD2 KO background. Importantly, due to the requirement of prior 2’-O-methylation of C₆₂ and C₆₃ for efficient THUMPD2-TRMT112-mediated methylation (**Fig. 4d-i**), in both these conditions, m^2G_{72} is reduced. Strikingly, the effects on snRNP incorporation and assembly are different when comparing loss of only m^2G_{72} (THUMPD2 KO) to the reduction of several 2’-O-methylations together with reduced/absent m^2G_{72} (LARP7 KD). Our results indicate that LARP7 KD significantly reduces U6 incorporation into snRNPs, whereas THUMPD2 KO leads to accumulation of U6 in di-/tri-snRNPs. It appears, therefore, that the 2’-O-methylations and m^2G_{72} in U6 have distinct functional roles. As the 2’-O-methylations seem necessary at an earlier stage of U6 snRNP assembly than m^2G_{72} , their effect predominates over the absence of m^2G in the combined LARP7 KD + THUMPD2 KO. These results are presented in **Fig. 6e-I** and described in the revised manuscript.

Reviewer: “11- I would like to see full-size gels of all Western and Northern blots as a supplementary file to ensure that the data is not selectively presented.”

Reply: Uncropped versions of all replicates of all western and northern blots are presented in the Source Data file provided with the revised manuscript.

Reviewer: “12- I appreciate the authors for providing a detailed materials and methods section.”

Reply: We thank the reviewer for this remark about the Materials and Methods section.

Minor comments

Reviewer: “1- All prime symbols are incorrect. An apostrophe is not a prime symbol, and please ignore my use of one in my review report for ease of writing.”

Reply: All the prime symbols in the manuscript and figures have been corrected.

Reviewer: “2- Line 118 – Citation for the METTL16 being the U6 snRNA m6A methyltransferase should include PMID: 28525753 and PMID: 29262316, and U6 snRNA A43 being involved in splicing should also cite PMID: 36409063 and PMID: 38808663.”

Reply: The missing references are now included.

Reviewer: “3- Line 147 – Is a punctuation mark missing before “so”?”

Reply: A comma has been included to separate the two independent clauses.

Reviewer: “4- In Figure 2e, the GFP panel shows that the cell on the middle left has an incorrectly placed arrow.”

Reply: The misplaced arrow has now been corrected.